# Astrocyte allocation during brain development is controlled by Tcf4-mediated fate restriction

Yandong Zhang [1,3], Dan Li[1,3], Yuqun Cai[1,3], Rui Zou[1], Yilan Zhang[1], Xin Deng [1], Yafei Wang [1], Tianxiang Tang[1], Yuanyuan Ma[1], Feizhen Wu [2] & Yunli Xie [1✉]

## Abstract

**Astrocytes in the brain exhibit regional heterogeneity contributing to regional circuits involved in higher-order brain functions, yet the mechanisms controlling their distribution remain unclear. Here, we show that the precise allocation of astrocytes to specific brain regions during development is achieved through transcription factor 4 (Tcf4)-mediated fate restriction based on their embryonic origin. Loss of Tcf4 in ventral telencephalic neural progenitor cells alters the fate of oligodendrocyte precursor cells to transient intermediate astrocyte precursor cells, resulting in mislocalized astrocytes in the dorsal neocortex. These ectopic astrocytes engage with neocortical neurons and acquire features reminiscent of dorsal neocortical astrocytes. Furthermore, Tcf4 functions as a suppressor of astrocyte fate during the differentiation of oligodendrocyte precursor cells derived from the ventral telencephalon, thereby restricting the fate to the oligodendrocyte lineage in the dorsal neocortex. Together, our findings highlight a previously unappreciated role for Tcf4 in regulating astrocyte allocation, offering additional insights into the mechanisms underlying neurodevelopmental disorders linked to Tcf4 mutations.**

**Keywords** Astrocyte Allocation; Transcription Factor 4; Fate-restriction; Gliogenesis; Pitt-Hopkins Syndrome
**Subject Categories** Development; Neuroscience

## Introduction

Astrocytes are the most abundant glial cells in the mammalian central nervous system and play essential roles in regulating synaptic activity (Allen, 2014; Chung et al, 2015; Tan and Eroglu, 2021), maintaining the integrity of the blood-brain barrier (Kaplan et al, 2020), and providing metabolic support to neurons (Weber and Barros, 2015). Although astrocytes are distributed throughout various regions of the brain, they exhibit regional differences in the morphological features and transcriptional profiles (Bayraktar et al,

2020; Ben Haim and Rowitch, 2017; Khakh and Deneen, 2019). This regional diversity is crucial in establishing specialized neuronglia units, which contributes to the regulation of local neural circuits that control specific behaviors (Huang et al, 2020; Kim et al, 2014; Morquette et al, 2015; Ribot et al, 2021), highlighting the importance of the astrocyte heterogeneity. Importantly, even after injury, astrocytes retain their regional adherence in the spinal cord (Tsai et al, 2012), suggesting that intrinsic positional cues play a vital role in regional astrocyte allocation. Alterations in astrocyte functions have been implicated in the progression of various neurological diseases, highlighting the potential for future therapeutic strategies targeting astrocytes (Brandebura et al, 2023; Herrero-Navarro et al, 2021; Lee et al, 2022). However, how astrocytes become diverse regionally remains unclear.

Astrocytes are generated from radial glial cells (RGCs) in the ventricular zone during brain development (Kriegstein and Alvarez-Buylla, 2009; Lin et al, 2021). In the dorsal neocortex, RGCs express the homeobox transcription factor Emx1 (Gorski et al, 2002) and undergo differentiation into glial progenitor cells towards the end of neurogenesis. These progenitor cells further differentiate into astrocytes and/or oligodendrocytes (Clavreul et al, 2022; Rowitch and Kriegstein, 2010). In vivo fate mapping of RGCs and their progeny shows that astrocytes are distributed within radial columns, confining them to the dorsal neocortex (Gao et al, 2014; Shen et al, 2021; Tsai et al, 2012). In the ventral telencephalon, astrocytes originate from RGCs present in both medial ganglionic eminence (MGE) and the preoptic area (POA), distinguished by their expression of the homeodomain transcription factor Nkx2.1 (Minocha et al, 2017; Torigoe et al, 2015). Meanwhile, RGCs expressing Nkx2.1 (Nkx2.1-RGCs) also give rise to interneurons and oligodendrocyte precursor cells (OPCs) (Fishell and Kepecs, 2020; Kessaris et al, 2006; Lim et al, 2018; Zhang et al, 2021a). While interneurons and OPCs migrate tangentially from the ventral telencephalon to the dorsal neocortex (Boda et al, 2022; Fishell and Kepecs, 2020; Kessaris et al, 2006; Yang et al, 2022), astrocytes remain in the ventral telencephalon strictly (Tsai et al, 2012). However, the mechanisms governing the regional allocation of astrocytes based on their embryonic origin remain unknown.

Astrocytes in the dorsal neocortex and the ventral telencephalon arise from distinct RGCs located within these spatially segregated

[1]Department of Anesthesia, State Key Laboratory of Medical Neurobiology and MOE Frontiers Center for Brain Science, Institutes of Brain Science, and Zhongshan Hospital, Fudan University, Shanghai 200032, China. [2]Laboratory of Epi-Informatics, Intelligent Medicine Institute of Fudan University, Shanghai 200032, China. [3]These authors contributed equally: Yandong Zhang, Dan Li, Yuqun Cai. ✉E-mail: yunli.xie@fudan.edu.cn

brain regions, making them a well-suited model system to investigate the mechanisms underlying regional astrocyte allocation based on their embryonic origin. In this study, we revealed that this process is achieved through cell fate restriction mediated by the transcription factor 4 (Tcf4), a basic helix-loop-helix (bHLH) transcription factor. Tcf4 often forms homo- or heterodimers with cell type-specific bHLH transcription factors, which recognize canonical E-box DNA elements and act as a transcription activator (Flora et al, 2007; Forrest, et al, 2014). Mutations in TCF4 cause Pitt-Hopkins syndrome, a neurodevelopmental disorder characterized by intellectual disability and autistic behavior (Li et al, 2019; Phan et al, 2020; Sweatt, 2013; Zhang et al, 2021b). We have previously demonstrated that Tcf4 plays crucial roles in regulating neuronal positioning during brain development (Zhang et al, 2021b). Building on our previous work in Tcf4 in the excitatory neuronal lineage (Wang et al, 2020b; Zhang et al, 2021b), we found that Tcf4 is highly expressed in the ventral telencephalon where Nkx2.1-RGCs located (Zhang et al, 2021a) (Appendix Fig. S1), suggesting that Tcf4 may play a role in the regulation of the Nkx2.1 lineage development. Here, we found that loss of Tcf4 in Nkx2.1-RGCs within the ventral telencephalon leads to misallocation of astrocytes in the dorsal neocortex. Despite their distinct origins, these astrocytes engage in interactions with neocortical neurons and blood vessels, exhibiting a similarity in morphological complexity and calcium activity when compared to the local astrocytes in the dorsal neocortex. Furthermore, single-cell sequencing showed that the depletion of Tcf4 in Nkx2.1-RGCs resulted in a fate transition of OPCs originating from the ventral telencephalon into transit intermediate progenitor cells, eventually differentiating into astrocytes in the dorsal neocortex. We found that Tcf4 acts as a repressor of astrocyte-lineage genes, thereby confining the fate of OPC derived from Nkx2.1-RGC to the oligodendrocyte lineage in the dorsal neocortex. These findings reveal that fate restriction plays a pivotal role in orchestrating the precise allocation of astrocytes to distinct brain regions, providing novel insights into the embryonic origin of regional astrocyte allocation and contributing to our understanding of neurodevelopmental disorders potentially associated with Tcf4 mutations.

## Results

### Astrocytes are allocated to distinct brain regions according to their embryonic origin

Astrocytes are abundant in both the pallium and subpallium. To determine their embryonic development origin, we initially traced the progeny of Emx1-expressing RGCs (Emx1-RGCs) in the dorsal neocortex or Nkx2.1-RGCs in the ventral telencephalon. We employed the tamoxifen-inducible Emx1-CreER$^{T2}$ or Nkx2.1-CreER$^{T2}$ system to activate the EYFP transgenic Cre reporter (Ai3) (Fig. EV1). Following tamoxifen induction during early embryonic stages and subsequent examination at postnatal stages, we observed that astrocytes, expressing the astrocytic marker Sox9 but not the oligodendrocyte marker Olig2, were distinctly allocated in the dorsal neocortex upon activation of Emx1-CreER$^{T2}$ (Fig. EV1A) or in the ventral telencephalon upon activation of Nkx2.1-CreER$^{T2}$ (Fig. EV1B), demonstrating that astrocytes are

regionally allocated according to their embryonic origin (Fig. EV1C) within spatially distinct regions of the brain.

### Tcf4 deletion impairs regional astrocyte allocation

To investigate the roles of Tcf4 in the allocation of astrocytes generated from Nkx2.1-RGCs in the ventral telencephalon, we used Nkx2.1-Cre driver to delete conditional alleles of Tcf4 (Tcf4$^{f/f}$; Nkx2.1-Cre, Tcf4 cKO), resulting in Tcf4 depletion in Nkx2.1-RGCs and their progeny. In addition, to visualize the Nkx2.1-RGC progeny, we crossed either Tcf4 cKO or wild-type (WT, Nkx2.1-Cre) mice with Ai3 reporter mice (Madisen et al, 2010). Depletion of Tcf4 in Nkx2.1-RGCs does not affect their proliferation and differentiation (Zhang et al, 2021a). When examined at P28, WT brains showed an absence of astrocytes derived from the Nkx2.1 lineage in the neocortex, as expected (Fig. 1A). However, in the neocortex of Tcf4 cKO brains, a substantial number of GFP-positive cells displaying astrocyte-like morphology were observed (Fig. 1B,C). When analyzed the distribution of mislocated astrocytes across cortical layers and found that these astrocytes were distributed throughout all cortical layers of the neocortex, with more astrocytes located in the deep layers than in the upper layers (Fig. 1D). In addition, we analyzed the distribution of mislocated astrocytes from the lateral to medial cortex and found that the number of mislocated astrocyte in the lateral cortex is higher than in the medial cortex (Fig. 1E). To determine whether these astrocytes arose from committed neuronal progenitors, we employed Dlx5/6-Cre to deplete Tcf4 in differentiating neuronal progenitors (Batista-Brito et al, 2008). No astrocytes were found in the neocortex of Tcf4$^{f/f}$; Dlx5/6-Cre mice (Fig. EV2A,B), indicating that they did not derive from committed neuronal progenitors. The depletion of Tcf4 was confirmed by in situ hybridization and real-time quantitative PCR (qPCR) (Fig. EV2C–F). The identity of astrocyte in the Tcf4 cKO neocortex was confirmed by the expression of the glial fibrillary acidic protein (GFAP) (Fig. 1F) and pan-astrocytic markers Sox9, S100b, and Aldh1l1 (Fig. 1G–J), while not expressing the pan-oligodendrocyte marker Olig2 (Fig. 1K), the mature oligodendrocyte marker CC1 (Fig. 1L) or the OPC marker PDGFRα (Fig. 1M). Notably, astrocytes originating from Nkx2.1-RGCs did not overlap with the local astrocytes derived from Emx1-RGCs in the dorsal neocortex (Fig. 1F, white arrow). When examining the development of astrocytes derived from Nkx2.1-RGCs in the neocortex of Tcf4 cKO brains (referred to as ectopic astrocytes), we found that cells with obvious astrocyte morphology were absent at P7 (Fig. 1N) but became apparent at P15, persisting in the neocortex even at P60 (Fig. 1N,O). Thus, Tcf4 plays a crucial role in preventing the allocation of astrocytes derived from RGCs of MGE/POA in the ventral telencephalon into the dorsal neocortex.

To determine whether the entire population of astrocytes in the dorsal neocortex, including ectopic astrocytes and local astrocytes, was affected by the loss of Tcf4, we stained the sections from both WT and Tcf4 cKO brains for the pan-astrocyte marker Aldh1l1. We found that the density of astrocytes in the neocortex was comparable between WT and Tcf4 cKO at both P7, when astrocytes are immature, and P21, when they are mature (Fig. EV3). This suggests that the astrocyte density might be rebalanced in Tcf4 cKO brains. In addition, we observed that the oligodendrocyte population remained unchanged in Tcf4 cKO brains compared to WT

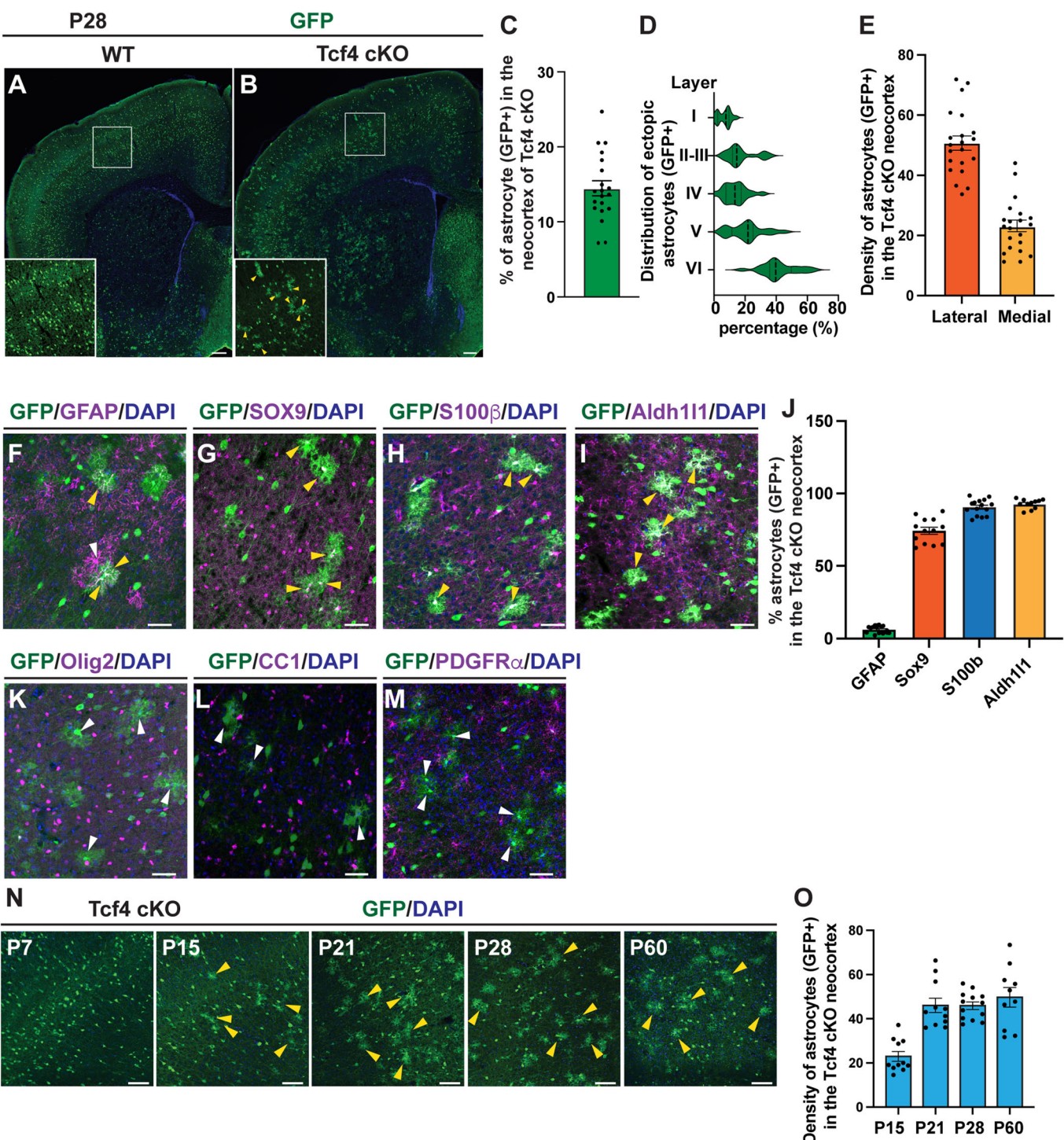

**Figure 1. Tcf4 is required to prevent the allocation of ventral astrocytes in the dorsal neocortex.**

(A, B) Immunostaining of GFP, a reporter for the Nkx2.1 lineage in WT (A) and Tcf4 cKO (B) brains at P28. Arrowheads indicate astrocyte-like cells in the neocortex of Tcf4 cKO brains. Scale bar, 200 μm. (C) Quantification of the percentage of astrocyte-like cells among all GFP + cells within the neocortex. Error bars show mean ± SEM (n = 3 mice). (D) The distribution of astrocyte-like cells across different layers of the neocortex in Tcf4 cKO brains. Error bars show mean ± SEM (n = 3 mice) (E) The density of astrocyte-like cells in the lateral and medial cortex in Tcf4 cKO brains. Error bars show mean ± SEM (n = 3 mice). (F–I) Co-immunostaining for GFP and astrocyte markers, (F) GFAP, (G) Sox9, (H) S100β, (I) Aldh1l1 in the neocortex of Tcf4 cKO brains. Yellow arrowheads indicate ectopic astrocyte and white arrowhead indicates local astrocytes. Scale bar, 50 μm. (J) Quantification of ectopic astrocytes expressing GFAP, Sox9, S100b, and Aldh1l1. Error bars show mean ± SEM (n = 3 mice for each experiment). (K–M) Co-immunostaining for GFP and (K) oligodendrocyte/OPC marker Olig2, (L) mature oligodendrocyte marker CC1, (M) OPC maker PDGFRα in the neocortex of Tcf4 cKO brains. Arrowheads indicate ectopic astrocytes do not express oligodendrocyte markers. Scale bar, 50 μm. (N) Representative images of ectopic astrocytes in different developmental stages. Yellow arrowheads indicate ectopic astrocytes. Scale bar, 50 μm. (O) Quantification of the density of ectopic astrocytes in different neocortical developmental stages. Error bars show mean ± SEM (n = 3 mice for each stage).

brains (Appendix Fig. S2). Oligodendrocytes can compensate for changes in their numbers from different origins (Kessaris et al, 2006; Foerster et al, 2024). Therefore, the loss of Tcf4 from the Nkx2.1 lineage did not impact the oligodendrocyte population in the dorsal neocortex. Consistent with this, we also observed that the number of astrocytes and oligodendrocytes derived from the Nkx2.1 lineage in the ventral telencephalon was not altered by the loss of Tcf4 (Appendix Fig. S3).

## Ectopic astrocytes exhibit dorsal-astrocyte-like characteristics

In the Tcf4 cKO neocortex, astrocytes derived from Emx1-RGCs of the dorsal neocortex remained unaffected as the depletion of Tcf4 occurred specifically in Nkx2.1-RGCs of the ventral telencephalon through cell type-specific Cre-mediated recombination (Zhang et al, 2021a). To investigate how these ectopic astrocytes interact with the local neurons and astrocytes within the dorsal neocortex, we performed *in utero* electroporation (IUE) to label Emx1-RGCs of the dorsal neocortex at E16.5 with plasmids expressing PB-GFP and PBase in the Tcf4 cKO background (Fig. 2A). After brain clearing using Clear, Unobstructed Brain/Body Imaging Cocktails and Computational analysis (CUBIC) (Susaki et al, 2015), both local astrocytes (green, derived from Emx1-RGCs of the dorsal neocortex) and ectopic astrocytes (red, derived from Nkx2.1-RGCs of ventral telencephalon lacking Tcf4) were simultaneously detected (Fig. 2B). When examining the morphological complexity of both local and ectopic astrocytes in the dorsal neocortex, we found that ectopic astrocytes displayed similar complexity to local astrocytes, with slightly increased volumes as revealed by 3D reconstructions (Fig. 2C–E). However, they differed from astrocytes in the ventral telencephalon (Fig. 2C–E), suggesting that ectopic astrocytes could be influenced by the local environment in the dorsal neocortex. Further analysis of the ratio of neurons to astrocytes, combined with NeuN staining, showed a similar neuronal investment in both local and ectopic astrocytes (Fig. 2F,G). This suggested that both local and ectopic astrocytes impinge upon a similar number of neuronal somata, even though ectopic astrocytes originated from RGCs of the ventral telencephalon. Meanwhile, we also found that these ectopic astrocytes exhibited direct vascular contact (Fig. 2H), a hallmark of cerebral astrocytes. To examine their functional properties, we investigated intracellular $Ca^{2+}$ transients, which are key features of astrocytes in response to their interactions with blood vessels and neurons (Paukert et al, 2014). Using Adeno-Associated viruses (AAVs) expressing the genetic calcium indicator GCaMP6f driven by an astrocyte-specific gfaABC1D promoter, we observed a strong response in ectopic astrocytes to the α1 adrenoceptor agonist phenylephrine (PE) application (Fig. 2I,J), with $Ca^{2+}$ signals similar to these evoked in local astrocytes (Fig. 2J,K). Thus, despite their different embryonic origins, ectopic astrocytes exhibit similar characteristics in the neocortex as local astrocytes.

## Tcf4 deletion in the Nkx2.1 lineage alters glial progenitor fate in the dorsal neocortex

To examine how ectopic astrocytes were generated in Tcf4 cKO brains, we performed single-cell RNA sequencing (scRNA-seq) on Nkx2.1-lineage cells isolated by Fluorescence-activated Cell Sorting

(FACS) from the neonatal cortices of WT and Tcf4 cKO brains at P8, a stage when glial progenitor cells undergo proliferation and differentiation (Ge et al, 2012) (Fig. 3A). After data integration and stringent quality control (Appendix Fig. S4A,B), we used unsupervised clustering in UMAP projection to identify 10 main clusters with distinct molecular features, including astrocyte precursor cells (APCs), oligodendrocyte precursor cells (OPCs), interneurons (INs) and endothelial cells (Fig. 3B). The signature genes of each cluster exhibited high specificity to different cell types (Fig. 3C). OPCs and INs were differentiated from Nkx2.1-RGCs of the ventral telencephalon, and migrated tangentially to the dorsal neocortex (Lim et al, 2018; Bandler et al, 2022). However, projection neurons derived from the dorsal neocortical RGCs were not found (Appendix Fig. S4C–F), confirming cells analyzed were of ventral origin. Notably, we identified a cell cluster expressing both astrocytic signature genes (e.g., Slc1a3, Aldh1l1, and Mfge8) and OPC genes (e.g., PDGFRα and Egr1) (Fig. 3B,C), suggesting the existence of transitional intermediate progenitor (TIP) cells. By comparing WT and Tcf4 cKO transcriptome, we evaluated how the loss of Tcf4 affected glial progenitor populations (Fig. 3D). Most notably, there was a significant decline in oligodendrocyte lineage representation (OPC1, OPC2 and immature oligodendrocyte (imOL) clusters) along with a concomitant increase in astrocyte progenitors (APC cluster) and TIP cells. TIP cells were present in Tcf4 cKO but not WT cortices (Fig. 3D,E), suggesting that the fate transition occurred upon the loss of Tcf4. TIP cells, similar to OPCs, expressed PDGFRα and Egr1 (Fig. 3F). However, they also exhibited high levels of APC markers, such as Mfge8 and Aldh1l1 (Fig. 3F). To further reconstruct the developmental trajectory of TIP cells, we performed pseudo-time analysis using Monocle3 (Qiu et al, 2017a) and identified the differentiation of funder cells (OPC1) into TIP cells (Fig. 3G). The pseudotemporal ordering of the astrocyte lineage revealed a simple early-to-late trajectory from TIP cells to APCs (Fig. 3H). Pseudotemporal gene expression analysis demonstrated a decrease in the expression of genes related to OPC development and an increase in astrocyte genes during the pseudotemporal progression trajectory (Fig. 3I,J). Overall, these early transcriptome changes paralleled the accompanying increase of TIP cells following Tcf4 loss, suggesting that ectopic astrocytes are derived from TIP cells in the Nkx2.1 lineage of dorsal Tcf4 cKO neocortex.

To validate the cell fate transition from OPCs to APCs in the Nkx2.1 lineage of Tcf4 cKO neocortex, we examined the changes in the number of OPCs and APC derived from Nkx2.1-RGCs in the dorsal neocortex. The analysis revealed a significant increase in the density of astrocyte progenitors (expressing BLBP, Aldh1l1) (Fig. 4A–D), and a decrease in the density of OPCs (expressing PDGFRα) (Fig. 4E,F) in the Nkx2.1 lineage in Tcf4 cKO brains compared to WT brains. During early cortical development, astrocyte progenitors proliferate to expand their number and can be labeled by replication-defective murine leukemia retroviruses expressing GFP (Ge et al, 2012). Accordingly, we locally injected viruses into the neocortex of both WT and Tcf4 cKO at P3 when progenitor cells are proliferating, and analyzed them at P21 when astrocytes are mature (Fig. EV4A). While only local astrocytes were labeled in the WT neocortex, ectopic astrocytes were observed in the neocortex of Tcf4 cKO (Fig. EV4B), suggesting a fate transition occurred upon the loss of Tcf4 at the time of viral infection. Together, these data demonstrate that Tcf4 suppresses astrocyte progenitor fate in the Nkx2.1 lineage of the dorsal neocortex.

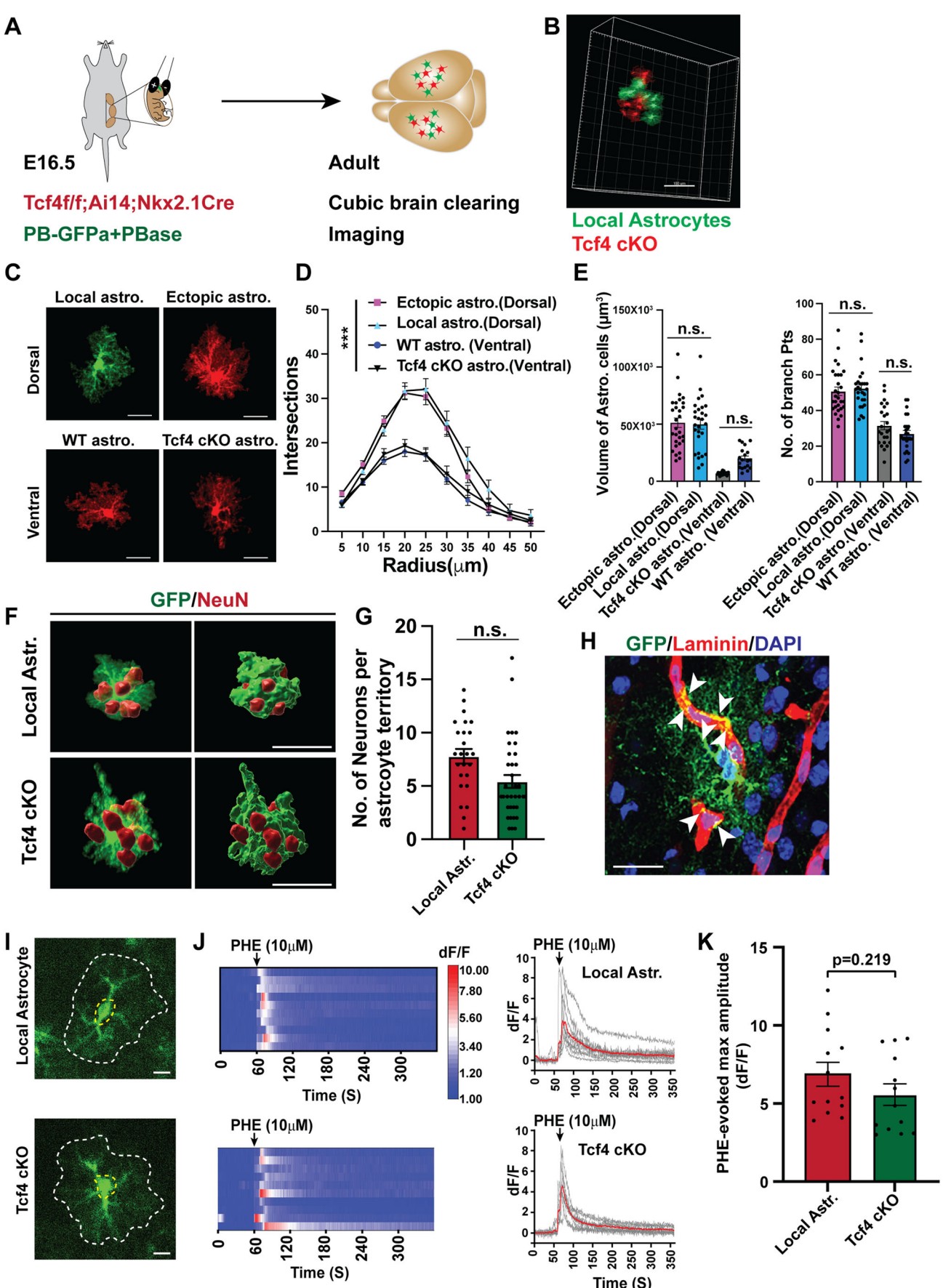

**Figure 2. Ectopic astrocytes share similar characteristics with local neocortical astrocytes.**

(A) Experimental design for labeling local neocortical astrocytes and ectopic astrocytes. (B) 3D-construction of the morphology of local astrocytes labeled with GFP (green) and ectopic astrocytes labeled with RFP (red) after tissue clearing. (C) Confocal volumes of GFP-positive local astrocytes, RFP-positive ectopic astrocytes in the dorsal neocortex, and RFP-positive astrocytes in the ventral brain from WT and Tcf4 cKO brains. Scale bar, 25 μm. (D) Sholl analysis of complexity of astrocytes derived from the Emx1 lineage (Local astro. Dorsal), ectopic astrocytes in the Tcf4 cKO neocortex (Ectopic astro. Dorsal), and astrocytes derived from the Nkx2.1 lineage in the ventral telencephalon in WT (WT astro. Ventral) and Tcf4 cKO (Tcf4 cKO astro. Ventral) brains ($n = 3$, cells from 3 mice per experiment per genotype were analyzed). Linear mixed model and ANOVA were used to calculate $P$-value, $P = 0.0000846$. Error bars show mean ± SEM. (E) Statistics of the volume and the number of branch points of astrocytes ($n = 3$, cells from 3 mice per experiment per genotype were analyzed). ANOVA followed by multiple comparisons was performed to determine the difference between every two groups using linear mixed models. Error bars show mean ± SEM. (F, G) Local and ectopic astrocytes (green) establish contacts with neurons (red). Scale bar, 50 μm. (G) Quantification of the number of neurons in a single astrocyte territory ($n = 25$ cells in at least 3 control brains, $n = 37$ cells in at least 3 mutant brains). (H) Ectopic astrocytes interact with blood vessels. White arrowheads indicate co-labeling signals of the ectopic astrocyte and the blood vessel. Scale bar, 50 μm. (I–K) Calcium activities of local and ectopic astrocytes induced by PHE. (I) Representative images of the calcium events in local and ectopic astrocytes. The nuclear territory and the astrocyte territory are circled by yellow and white dashed lines. Scale bar, 10 μm. (J) Heatmaps and traces of calcium fluctuations of local (top) and ectopic astrocytes (bottom) before and after PHE induction. Average activities are showed as red line. (K) Quantification of PHE-evoked maximum amplitudes in local and ectopic astrocytes ($n = 4$ mice per group). All quantifications are resulted from at least 3 brains for each phenotype. T-test was performed to compare the two groups using linear mixed models. n.s. non-significant. ***$P < 0.001$. Error bars represent the mean ± SEM.

## Activation of astrocyte genes by increasing chromatin accessibilities in enhancers promotes cell fate transition upon the loss of Tcf4

To investigate the mechanisms underlying the fate transition from OPCs to APCs, we reanalyzed the clusters of OPCs and APCs (Appendix Fig. S5A). Pseudotime-analysis-derived developmental trajectory analysis showed that TIP cells were differentiated from OPCs (Fig. 5A), consistent with OPC1 being the major founder cell (Fig. 3G). Particularly, genes expressed in oligodendrocyte lineage regulating myelination (e.g., Sox10, Mbp, and Olig1) were down-regulated, while genes related to astrocyte differentiation (e.g., S100b, Stat3, NFIB, and NFIX) were increased during the developmental trajectory from OPCs to TIP cells (Fig. 5B), supporting the hypothetical fate transition from OPC to APC and suggesting transcriptional changes toward an astrocyte fate is activated in the Nkx2.1 lineage of the dorsal neocortex upon the loss of Tcf4. To elucidate changes in chromatin landscape during the transition from OPC to APC, we performed assay for transposase-accessible chromatin using sequencing (ATAC-Seq) in a lineage-specific manner on sorted cells from the Nkx2.1 lineage of the dorsal neocortex (Fig. 5C–E; Appendix Fig. S5B,C). A total of 32,120 consensus peaks were recovered using DiffBind and 66.25% of these peaks overlapped with those identified in a reference analysis of brain tissue using DNase-Seq from the ENCODE database (Data ref: Encyclopedia of DNA Elements ENCSR000COG, 2012) (Appendix Fig. S5D). Differential analysis of ATAC peaks revealed prominent changes (1129 closing regions, 777 opening regions) in chromatin accessibility upon the loss of Tcf4 (Fig. 5E,F), with a majority of the peaks located in intergenic regions and introns (Appendix Fig. S5E). Functional annotation of peak-associated genes using Genomic Regions Enrichment of Annotations Tool (GREAT) (McLean et al, 2010) highlighted that closing chromatin regions were associated with genes related to oligodendrocyte differentiation and myelination (Olig2, Myrf, Tcf7l2) while opening chromatin regions were linked to genes associated with astrocyte development (Fgfr3, NFIB, NFIX) (Fig. 5G). Motif enrichment analysis in differentially accessible regions revealed enriched binding motifs for bHLH- and Sox-related transcription factors in closing chromatin regions, while astrogenic NFI- and Sox-related transcription factors in opening chromatin regions in Tcf4 cKO brains (Fig. 5H). Thus, these

findings suggest that the loss of Tcf4 in the Nkx2.1 lineage leads to alterations in chromatin accessibility, promoting the activation of astrocyte-related genes and the suppression of OPC-related genes.

To examine changes in histone modifications associated with transcriptional enhancers (H3K4me1), active enhancers and promoters (H3K27ac) and polycomb factor repression (H3K27me3), we performed Cleavage Under Targets and Tagmentation (Cut&Tag) (Fig. 6A; Appendix Fig. S6A–F). A significant overlap of 89.93% was observed between ATAC peaks and H3K4me1 peaks (Appendix Fig. S6G), suggesting that a considerable proportion of distal ATAC peaks may function as potential enhancers. To further classify cis-regulatory elements associated with ATAC peaks, we integrated published regulatory regions from the ENCODE3 enhancer gene map, H3K4me1 peaks generated in this study, and regulatory regions from the Ensemble Regulatory Build (Zerbino et al, 2015) as a reference dataset. This reference dataset classified different regions into promoters, enhancers, and other distal elements (Appendix Fig. S6H,I). Upon overlapping our ATAC consensus peaks with this reference dataset, 13,909 peaks were assigned as promoters, while 15616 peaks were identified as enhancers (Appendix Fig. S6J). Notably, 84.59% of peaks within the closing regions (Fig. 6B) and 78.12% of peaks in the opening regions were identified as enhancers (Fig. 6C).

We next aimed to identify transcription factors that may drive changes in the chromatin landscape. Footprint analysis was further performed to uncover the dynamic binding activity of transcription factors during the fate transition (Bentsen et al, 2020). The results showed that a shift of bHLH-factor (e.g., Olig2 and Tcf12) activity in WT to activation of NFI (e.g., NFIA and NFIX) motifs in Tcf4 cKO, suggesting a change in transcriptional regulation upon the loss of Tcf4 (Fig. 6D). Given NFIA and NFIB promote astrogenesis (Canals et al, 2018; Deneen et al, 2006), we analyzed the expression of the whole NFI family genes in the pseudo-trajectory from OPC to TIP cells and found that their expression was increased (Fig. 6E), suggesting that NFI genes may be the key driver for astrogenesis in the Nkx2.1 lineage of Tcf4 cKO brains. Indeed, when profiling the binding activity of NFIX within the closing or opening chromatin regions, we found that the NFI transcription factor NFIX has higher binding activity in the opening regions compared to the closed regions (Fig. 6F). Consistently, when profiling the active histone mark H3K27ac and the polycomb-repressive histone mark H3K27me3, we found that the NFI transcription factor NFIX was activated through

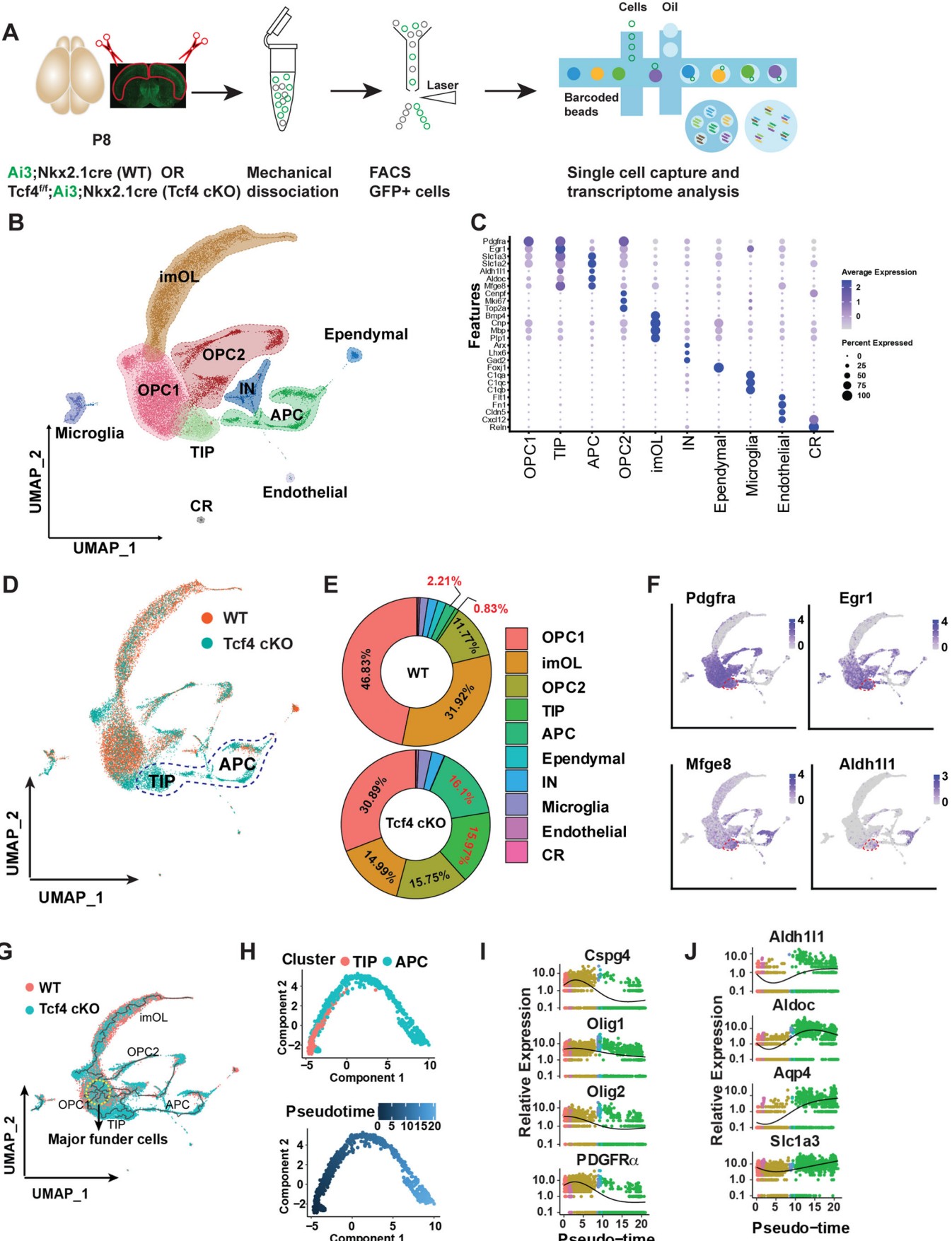

**Figure 3.  Single-cell RNA sequencing reveals transient intermediate progenitors in the Nkx2.1 lineage upon the loss of Tcf4.**

(A) A schematic illustrates the sample collection and single-cell RNA sequencing workflow for analyzing GFP-positive cells derived from the Nkx2.1 lineage in the neocortex of WT and Tcf4 cKO brains. (B) UMAP clustering of integrated cells from WT and Tcf4 cKO neocortices, with annotations for cell clusters based on transcriptomic signatures (4 brains of each group were pooled together for FACS and single-cell sequencing). (C) A dot plot representing the expression level of markers of each cluster. (D) UMAP visualization showing the distribution of cells of WT or Tcf4 cKO brains. (E) The proportion of distinct clusters from WT or Tcf4 cKO brains. (F) Expression of marker genes in the feature plot defines transient intermediate progenitors (TIP) that exist in Tcf4 cKO brains. (G) Global lineage progression is inferred by Monocle3, where the yellow-dashed circle indicates potential founder cells. (H) Pseudotemporal trajectory of the TIP and APC lineage. (I, J) The expression of oligodendrocyte (I) or astrocyte markers (J) along pseudo-time in (H).

enhancer-mediated regulation, while the myelin regulatory factor (Myrf), a transcription factor involved in oligodendrocyte development (McKenzie et al, 2014; Xiao et al, 2016), was inactivated (Fig. 6G,H). Moreover, when NFIX was overexpressed in neural progenitors through in utero electroporation, the number of astrocytes significantly increased (Fig. 6I, from 11.62% in control to 35.69% with NFIX overexpression). Taken together, these data demonstrate that the activation of genes related to astrogenesis, facilitated by changes in enhancer epigenetic states, promotes the fate transition from OPC to APC upon the loss of Tcf4.

## Tcf4 orchestrates chromatin accessibility to control glial cell fate

To investigate how Tcf4 regulates the expression of genes associated with astrocyte development in the Nkx2.1 lineage of the dorsal neocortex, we generated a lineage-specific expression of Flag/HA-tagged Tcf4 mouse line (Appendix Fig. S7A–D). In this model, we inserted the Flag/HA-tag at the C-terminus of the endogenous Tcf4 locus, allowing specific Flag/HA-tag expression within the Nkx2.1 lineage upon Nkx2.1-Cre activation (Appendix Fig. S7E). By performing Cut&Tag with an anti-Flag antibody on FACS-sorted Nkx2.1 lineage cells, we analyzed Tcf4 occupancy in a lineage-specific manner (Fig. 7A). After peak calling, we identified a high number (38662) of Tcf4 binding sites (Fig. 7B,C). A de novo search for enriched DNA motifs by Homer revealed the canonical E-box motif of Tcf4 (Fig. 7D). Notably, Tcf4 binding peaks were preferentially located in H3K4me1-enriched enhancer regions (Fig. 7E). We then analyzed Tcf4 binding activity in differentially accessible regions revealed by ATAC-seq (Fig. 5F) and found that the binding signal of Tcf4 was predominantly enriched in the closing chromatin regions (Fig. 7F), indicating that Tcf4 binding is required to maintain chromatin accessible at these loci. As the bHLH motifs were found in the closing regions (Fig. 5H), Tcf4 may directly bind to these regions to maintain the opening status of the chromatin and activate the gene expression to promote OPC differentiation.

The polycomb repressive complex 2 (PRC2) is essential for catalyzing methylation of H3K27 and represses gene expression during development (Laugesen et al, 2019; Pereira et al, 2010). We found that the enrichment of H3K27me3 was significantly decreased at the putative enhancers of NFIX after Tcf4 loss (Fig. 7G). Meanwhile, these regions also possess binding signals of Tcf4, Ezh2, and Suz12, components of the PRC2 complex (Fig. 7G), implicating that Tcf4 may associate with PRC2 complex to repress astrocytic genes. Indeed, using co-immunoprecipitation assay, we found Tcf4 was associated with PRC2 component Ezh2 (Fig. 7H). Taken together, these data demonstrate that Tcf4 mediated transcriptional program actively promotes the oligodendrocyte fate while repressing the undesired astrocyte fate, likely associated with

PRC2, in the Nkx2.1 lineage of the dorsal neocortex. Upon the loss of Tcf4, astrogenic genes are activated, which promotes the fate transition from the OPCs to APCs, indicating an essential role for Tcf4 in regulating the fate restriction of OPCs.

## Discussion

Regionally allocated astrocytes interact with neurons to establish local neural circuits (Allen et al, 2022; Cheng et al, 2023; Huang et al, 2020; Kofuji and Araque, 2021). However, how astrocytes are allocated to restricted brain regions spatially remains largely unknown. Here, we report that cell fate restriction governs the regional allocation of astrocytes based on their embryonic origin during brain development. Astrocytes are regional allocated, whereas OPCs within the cortex are derived from RGCs located in three distinct regions, including MGE, lateral and caudal ganglionic eminences (LGE-CGEs), and dorsal ventricular zone of the neocortex, following a temporal pattern during brain development (Kessaris et al, 2006). We propose a novel mechanism by which the restricted fate of OPCs originating from the ventral telencephalon ensures the specific allocation of astrocytes within this region. Tcf4 is required to retain the allocation of astrocytes derived from Nkx2.1-RGCs to the ventral telencephalon. While oligodendrocytes migrate tangentially from the ventral telencephalon to the dorsal neocortex, astrocytes generated from the same progenitors within the ventricular zone of the ventral telencephalon are exclusively allocated to the local region. To ensure proper regional allocation, Tcf4-mediated transcriptional regulation restricts the fate of OPCs produced by Nkx2.1-RGCs in the dorsal neocortex to the oligodendrocyte lineage but not the undesired astrocyte lineage, preventing ectopic allocation of astrocytes originated from Nkx2.1-RGCs in the dorsal neocortex. Previous study showed that astrocytes are restricted to specific domains of the spinal cord even after acute injury (Tsai et al, 2012), suggesting that the intrinsic properties of astrocytes in maintaining their spatial identity. Together with our data, we propose that the intrinsic transcriptional program plays an essential role in confining astrocyte allocation to restricted spatial domains according to their embryonic origin during brain development. We observed that ectopic astrocytes exhibit similarities with local astrocytes in morphology, interaction with blood vessels, and calcium activities, while differ from the ventral astrocytes derived from the same RGCs expressing Nkx2.1, suggesting that the local environment may influence the characteristics of ectopic astrocytes and prompt them to adopt local neural circuits.

During early cortical development, OPCs originating from Nkx2.1-RGCs in the ventral telencephalon migrate to the neocortex and differentiate into oligodendrocytes (Kessaris et al, 2006; van

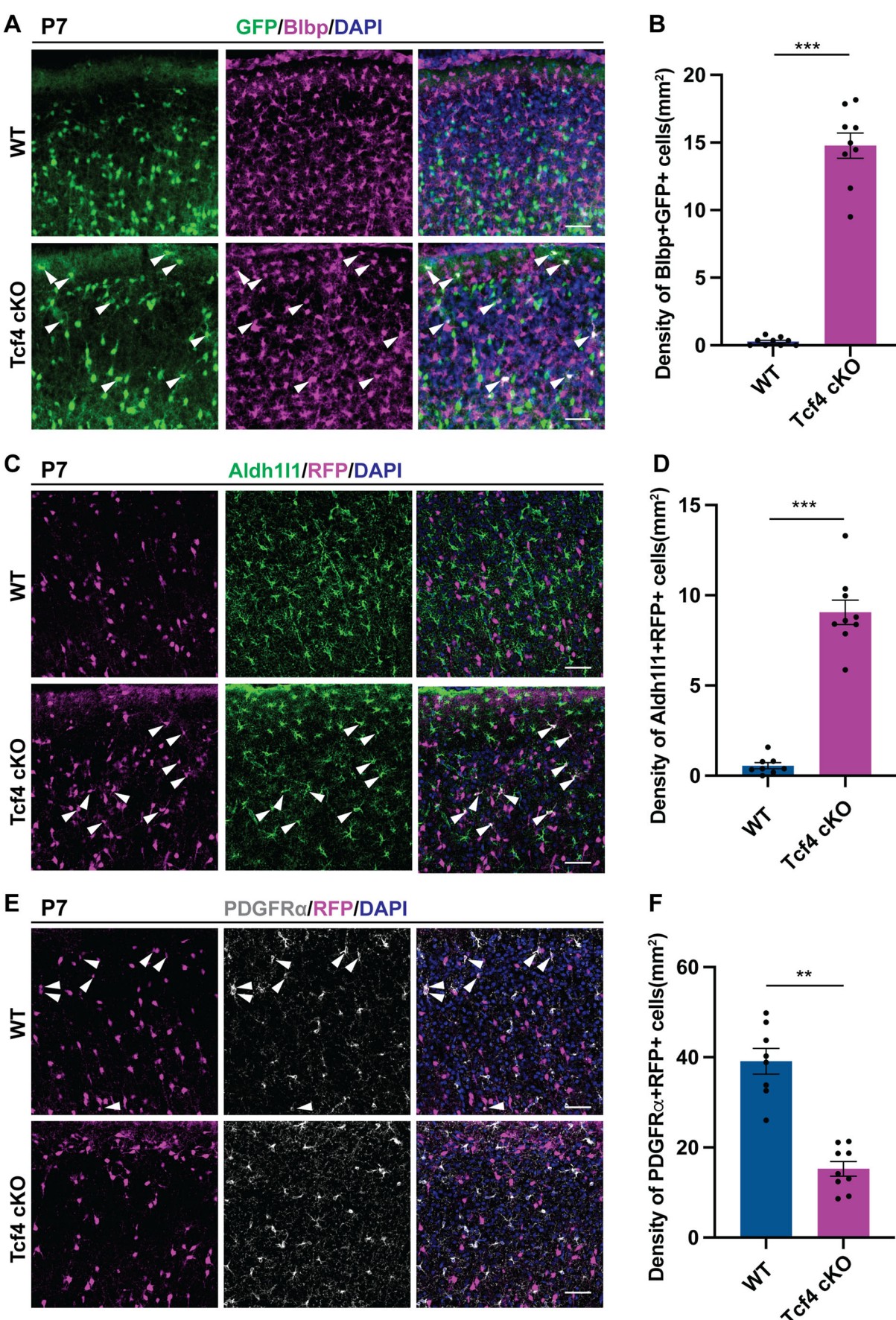

**Figure 4. Increase of glial progenitors expressing astrocyte marker genes in Tcf4 cKO brains.**

(A) Immunolabeling of GFP and Blbp in the neocortex of WT and Tcf4 cKO brains at P7. White arrowheads indicate glial progenitors expressing Blbp. Scale bar, 50 μm. (B) Quantification of the density of glial progenitors expressing Blbp ($n = 3$ mice for each genotype were analyzed), $P = 0.000113$. (C) Immunolabeling of RFP and Aldh1l1 in the neocortex of WT and Tcf4 cKO brains at P7. White arrowheads indicate glial progenitors expressing Aldh1l1. Scale bar, 50 μm. (D) Quantification of the density of Aldh1l1-expressing glial progenitors ($n = 3$ mice for each genotype were analyzed), $P = 0.000397$. (E) Immunolabeling of GFP and PDGFRα in the neocortex of WT and Tcf4 cKO brains at P7. White arrowheads indicate OPCs expressing PDGFRα. Scale bar, 50 μm. (F) Quantification of the density of OPCs expressing PDGFRα ($n = 3$ mice for each genotype were analyzed), $P = 0.005407$. T-test using linear mixed model is used for comparing two groups. Error bars represent mean ± SEM. ***$p < 0.001$, **$p < 0.01$.

Bruggen et al, 2022). Unlike APCs, OPCs proliferate and differentiate into OLs throughout the entire lifespan (Huang et al, 2013; Hill et al, 2024). In the early developmental stages, APCs and OPCs share some marker gene expression, such as NFI proteins and FGFR3 (Molofsky et al, 2012). The fate of OPCs is regulated by both intrinsic and extrinsic factors (Elbaz and Popko, 2019; He and Lu, 2013). However, the mechanisms that restrict OPC fate to prevent the generation of APCs remain unclear. Analysis of single-cell transcriptome from the Nkx2.1 lineage of Tcf4 cKO neocortex reveals a distinct population of cells expressing both OPC genes and APC genes, identified as TIP cells. These TIP cells subsequently undergo differentiation into astrocytes in the dorsal neocortex. These findings uncover the crucial role of Tcf4 in restricting the fate of OPCs in the Nkx2.1 lineage, preventing astrocyte development in the dorsal neocortex. ATAC-seq and Cut&Tag data demonstrate the prominent effect of Tcf4 loss on genes related to oligodendrocyte development, providing compelling evidence for Tcf4's role in promoting opening and active chromatin environments at these sites to restrict OPC fate in the Nkx2.1 lineage within the dorsal neocortex. Meanwhile, analysis reveals that the NFIX locus contains Tcf4-occupied sites in Nkx2.1 lineage cells, and this locus becomes more accessible upon the loss of Tcf4, promoting astrocyte development. Therefore, Tcf4 may play a lineage-specific role in controlling cell fate. This could be achieved by interacting with specific binding partners to regulate lineage-specific gene expression. The expression of Tcf4 in the Nkx2.1 lineage of the dorsal neocortex is essential for restricting OPC fate towards oligodendrocyte differentiation, but not astrocyte, which ultimately enables astrocytes to be regionally allocated according to their embryonic origin, highlighting the importance of the intrinsic program in regulating the restriction of OPC fate.

The dorsal neocortex and ventral telencephalon are distinct brain regions that are spatially separate and perform different functions in advanced behaviors. Astrocytes within these regions provide functional supports for neurons unique to each region (Cheng et al, 2023; Khakh and Deneen, 2019). However, a systemic analysis of the impact of ectopic astrocytes on local neuronal and glial functional networks in future studies would provide further insights into how regionally allocated astrocytes influence regional neuronal circuits. In addition, it is important to explore how these astrocytes contribute to mouse behaviors relevant to Pitt-Hopkins syndrome. Our findings reveal an unappreciated role of Tcf4 in determining the cell fate restriction of the Nkx2.1 lineage in the ventral telencephalon, which ensures that astrocytes are allocated to restrict regions based on their embryonic origin. This discovery holds the potential for generating region-specific astrocytes and provides further insights into the neurodevelopmental defects of Pitt-Hopkins syndrome, which is caused by mutations in TCF4.

# Methods

### Reagents and tools table

| Reagents/resource | Reference or source | Identifier or catalog number |
|---|---|---|
| **Experimental models** | | |
| Mouse: Tcf4[f/f] | Wang et al (2020b) | N/A |
| Mouse: Nkx2.1-Cre | Xu et al (2008) | N/A |
| Mouse: Tcf4-3xHA-3xFlag | This paper | N/A |
| **Recombinant DNA** | | |
| PB-GFP | This paper | N/A |
| PB-RFP | This paper | N/A |
| PBase | This paper | N/A |
| **Antibodies** | | |
| Chicken anti-GFP | Abcam | Ab13970 |
| Rabbit anti-GFAP | Dako | Z0334 |
| Goat anti-SOX9 | R&D Systems | AF3075 |
| Rabbit anti-S100β | Dako | Z0311 |
| Rabbit anti-Aldh1l1 | Abcam | Ab87117 |
| Rabbit anti-Olig2 | Millipore | AB9610 |
| Mouse anti-CC1 | Millipore | OP80-100UG |
| Rat anti-PDGFRα | BD Bioscience | 558774 |
| Rabbit anti-NeuN | Abways | CY5515 |
| Rabbit anti-Laminin | Sigma | L9393 |
| Rabbit anti-BLBP | Abcam | Ab32423 |
| Rabbit anti-RFP | Clontech | 632496 |
| Rabbit anti-HA | Cell Signaling Technology | 3724s |
| Mouse anti-Flag | Smart-Lifesciences | SLAB0101 |
| Rabbit anti-H3K4me1 | Abcam | ab8895 |
| Rabbit anti-H3K27me3 | Cell Signaling Technology | 9733s |
| Rabbit anti-H3K27ac | Abcam | ab4729 |
| Goat anti-Rabbit IgG H&L | Abcam | ab6702 |
| Goat anti-Mouse IgG H&L | Abcam | ab6708 |
| **Chemicals, enzymes and other reagents** | | |
| rAAV-pZac.1-GfaAGC1D-cyto-Gcam6f-SV40-PA | BrainVTA | PT1450 |
| DMEM | Thermo Scientific | 21063029 |
| HBSS | Thermo Scientific | 14025076 |
| protease inhibitors | Pierce | A32963 |

| Reagents/resource | Reference or source | Identifier or catalog number |
|---|---|---|
| Tamoxifen | Sigma | T6548 |
| Neural Tissue Dissociation Kit | Miltenyi Biotec | 130-092-628 |
| Hyperactive In-Situ ChIP Library Prep Kit for Illumina (pG-Tn5) | Vazyme | TD901 |
| TruePrep DNA Library Prep Kit V2 for Illumina | Vazyme | TD501 |
| TruePrep Index Kit V2 for Illumina | Vazyme | TD202 |
| MinEluteGel Extraction kit | Qiagen | 28606 |
| SPRIselect beads | Beckman Coulter | B23319 |
| MaXtract High Density Tubes | Qiagen | 129046 |
| DNA Clean Beads | Vazyme | N411 |
| **Software** | | |
| Imaris10.0.2 | https://imaris.oxinst.com | RRID:SCR_007370 |
| Seurat v3 | https://satijalab.org/seurat/ | RRID:SCR_016341 |
| MACS2 2.2.7.1 | https://github.com/macs3-project/MACS/wiki/Install-macs2 | N/A |
| Homer v4.11.1 | http://homer.ucsd.edu/homer/motif/ | RRID:SCR_010881 |
| Diffbind2 | https://bioconductor.org/packages/release/bioc/html/DiffBind.html | RRID:SCR_012918 |
| TOBIAS | https://github.com/loosolab/TOBIAS | N/A |
| Fiji/ImaeJ | https://imagej.net/software/fiji/downloads | RRID:SCR_002285 |
| Deeptools v3.5.1 | https://test-argparse-readoc.readthedocs.io/en/latest/index.html | RRID:SCR_016366 |

## Methods and protocols

### Experimental animals

Tcf4f/f (Wang et al, 2020b) mice were crossed to Nkx2.1-Cre (Xu et al, 2008) to generate Tcf4f/+; Nkx2.1-Cre mice, which were further crossed to Tcf4f/f to generate Tcf4f/f; Nkx2.1-Cre (Tcf4 cKO) mice. Tcf4-3xHA-3xflag mice were generated at BIOCYTO-GENE. All animal work was approved by Animal Care and Use Committee of Shanghai Medical College of Fudan University (20200306-12).

### Immunohistochemistry

For postnatal brain samples, animals were first anesthetized and perfused with 1x phosphate buffered saline (1x PBS) and followed by 4% PFA. Then brains were collected and fixed in 4% Paraformaldehyde (PFA) overnight except in the case of PDGFRα immunostaining where fixation for less than 4 h is necessary. Fixed brains were first washed with 1x PBS and dehydrated in 30%

sucrose dissolved in 1x PBS till they sank to the bottom. Brain samples were sectioned into 30 μm slices for immunostaining. For co-staining of Aldh1l1, PDGFRα, and GFP, antigens were retrieved in pH 6.0 citrate buffer at 65 °C for 5 min to protect PDGFRα and GFP signals. Slices were permeabilized in 0.5% Triton-X in 1x PBS for 30 min and then incubated in blocking buffer (5% donkey serum with 0.1% Triton-X in 1x PBS) for 1 h. After that, slices were incubated in primary antibody solution in 4 °C overnight. In the following day, slices were first washed with 1x PBS and then incubated in secondary antibody solution for 2 h at room temperature before mounting.

### Tamoxifen induction

Tamoxifen was dissolved in corn oil (20 mg/ml) by rotating overnight at room temperature. Timely pregnant females received a single intraperitoneal injection of tamoxifen (100 mg/kg of body weight) at E12.5. Live embryos were recovered at E19 through cesarean section, and raised until P21 for analysis.

### In utero electroporation

To label local astrocytes in the neocortex, we performed in utero electroporation on timed pregnant mice at embryonic day 16.5 (E16.5). After anesthetizing the pregnant mice, the embryos were carefully exposed. Plasmids expressing PBase and PB-GFP, diluted to a final concentration of 1 μg/μl in 1x PBS, were injected into the embryos' lateral ventricles. Electroporation was performed using a BTX system, delivering five 50 ms pulses of 35 V with 950 ms intervals. Upon completion of the procedure, the embryos were carefully returned to their original positions, and the surgical incisions were sutured. Post-surgery, the mice were returned to their home cages, allowing the electroporated embryos to develop normally until birth.

### Brain slice clearing using CUBIC

To analyze the morphology of ectopic astrocytes in the neocortex, tissue clearing was performed using CUBIC with slight changes (Susaki et al, 2015). First, after deep anesthetization of mice, cardiac perfusion was performed with 10 ml of 1x PBS and followed with 10 ml of 4% PFA. Brains were collected and post-fixed in 4% PFA at 4 °C overnight. Brains were sectioned into 400 to 500 μm slices. After washing the slices with 1x PBS for 2 h twice at room temperature, the slices were immersed into 1/2-water-diluted reagent1 at 37 °C on a shaker for 6–12 h. Then 1/2-water-diluted reagent was replaced with reagent1 and incubated at 37 °C on a shaker for 2–4 days till slices became transparent in reagent1. Then transparent slices were washed with 1x PBS for 2 h twice at room temperature and immersed in 20% sucrose in PBS at 4 °C till the slices sank into the bottom. After that, slices were washed with 1x PBS three times, 5 min per time slightly. Then slices were incubated in primary antibody solution, diluted in 0.1% (v/v) Triton-X-100, 0.5% (w/v) BSA, and 0.1% ProClean 300 in 1x PBS, for 8 days at room temperature. After incubation of the primary antibody, slices were washed with 5 ml of 0.1% (v/v) Triton-X-100 in 1x PBS for 2 h twice and incubated in secondary antibody solution diluted with the same buffer as primary antibody solution at room temperature for 8 days in the dark. Slices were washed with 5 ml of 0.1% (v/v) Triton-X-100 in 1x PBS for 3 times, and 2 h per time. After washing, slices were immersed in 1/2-PBS-diluted reagent2 and incubated at 37 °C overnight. Then 1/2-PBS-diluted

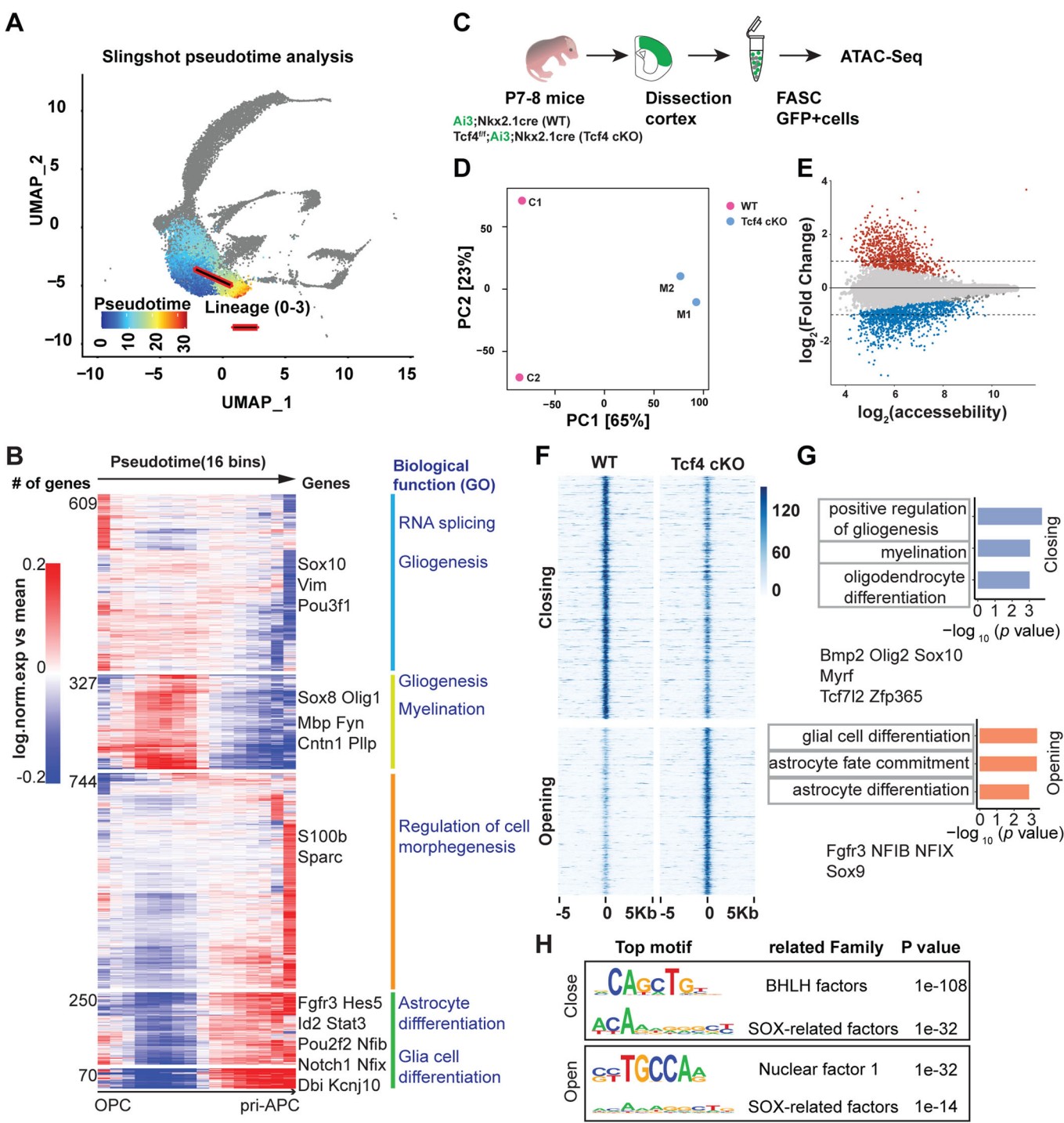

**Figure 5. Activation of astrocyte gene expression leads to a cell fate transition.**

(A) Pseudotime trajectory via slingshot analysis illustrates pseudo-progression from the OPC progenitor cluster 0 to TIP cluster 3. (B) Temporally expressed genes along the pseudotime trajectory in (A) and related biological functions by gene oncology (GO). The heatmap displays the mean relative expression of each gene in all cells in each pseudotime bin. (C) Schematic diagram for analyzing the chromatin accessibility of cells from the Nkx2.1 lineage in the dorsal neocortex. (D) Principal-component analysis (PCA) of ATAC-seq of GFP-positive cells from Nxk2.1 lineage in the neocortex of WT and Tcf4 cKO mice. (E) MA plot displaying differential accessible peaks (DA peaks) following deletion of Tcf4 in Nkx2.1 lineage. Each dot represents a peak, colored dots indicate differentially accessible peaks (FDR < 0.05 | log2(fold change) > 0.5). Closing regions, blue; opening regions, red. (F, G) Differentially accessible chromatin regions and related functions of genes. (F) Heatmaps of ATAC-seq signal plots over differentially accessible regions concentrated by peak center. (G) Bar plots show region-related GO terms annotated by GREAT, the P-value were calculated using hypergeometric test and binomial test defined in GREAT. (H) Transcription factor binding motif enriched in differentially accessible regions in (F), identified by de novo motif analysis using Homer, the P-values were calculated by Homer to indicate the enrichment of motifs in target sequence compared to background sequence.

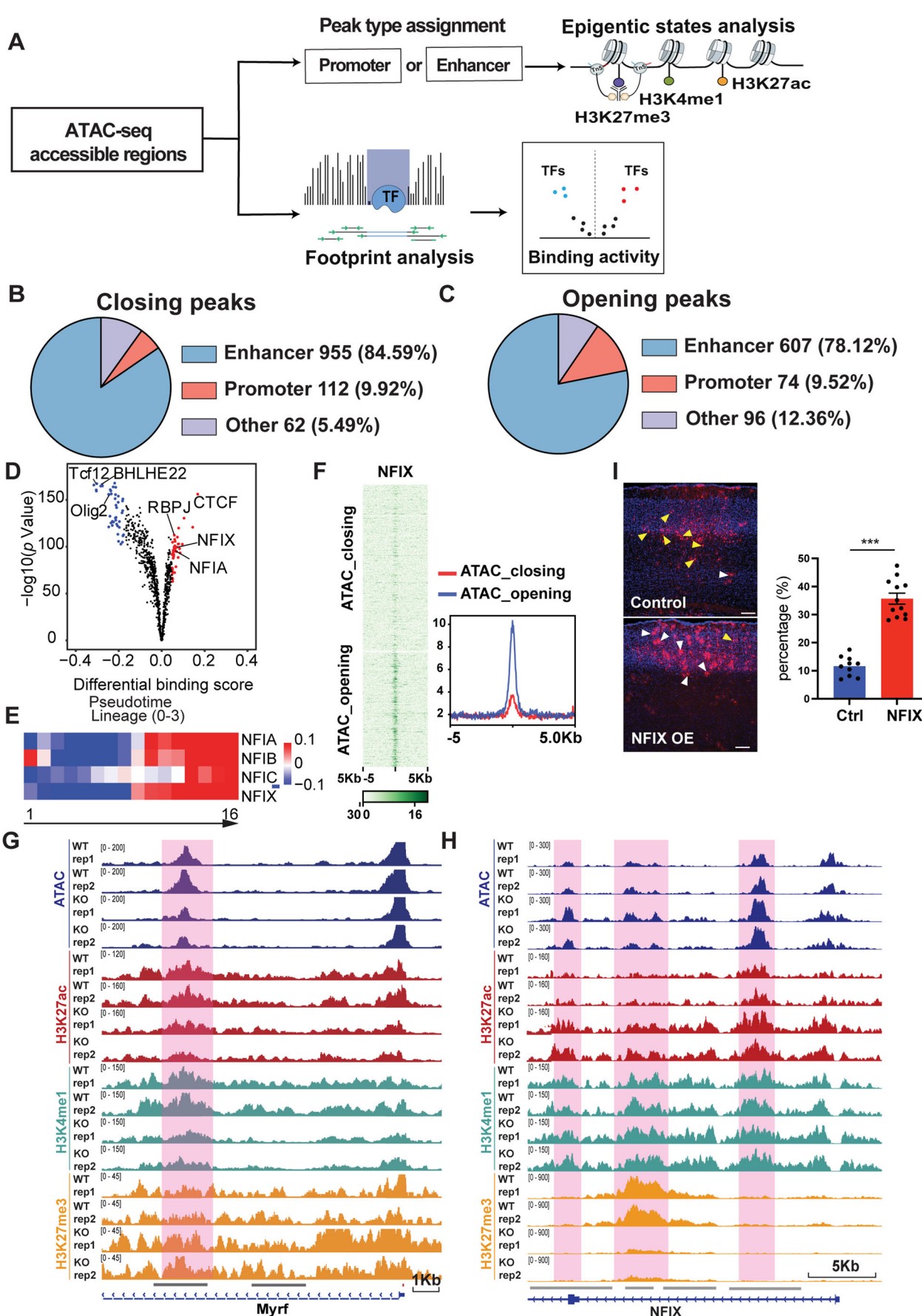

**Figure 6.  Activation of astrocyte genes by increasing chromatin accessibilities in enhancers promotes cell fate transition upon the loss of Tcf4.**

(A) A schematic diagram illustrates the integrated analysis strategy for ATAC seq data and Cut&Tag for histone modifications. ATAC peaks were assigned as promoters and enhancers according to the overlapping with reference datasets and their states were further confirmed by overlapping with histone markers. Footprint analysis was performed to examine the changes in the binding activity of transcription factors. (B, C) The peak assignment as promoters or enhancers among the closing peaks of ATAC (B), or among the opening peaks of ATAC (C). (D) Comparison of TFs binding activities in the Nkx2.1 lineage between WT and Tcf4 cKO brains. Volcano plots show the TOBIAS differential binding score on the x-axis and $-\log_{10}$ (p-value) on the y-axis, with each dot representing one TF, the P-value for each TF was calculated by comparing the enrichment from the target sequence to the background sequence in TOBIAS. The top 5% TFs are colored red in Tcf4 cKO and blue in WT. (E) The expression of NFI family transcription factors along pseudotime of the lineage (0–3). (F) Heatmap and profiles showed the binding activity of NFIX over differentially accessible regions centered by peak center. (G, H) Track snapshots illustrate chromatin accessibility and histone modifications in the predicted enhancer regions of Myrf (G) and NFIX (H). Gray boxes indicate predicted enhancers of embryonic mouse brains from ENCODE. (I) Representative images of control and NFIX-overexpression (NFIX-OE) brain sections at P7, and quantification of percentage of astrocytes in all RFP+ cells ($n = 3$ mice for each genotype were analyzed), $P = 2.4e{-}09$. T-test using linear mixed model is used for comparing two groups. Error bars represent mean ± SEM. ***$p < 0.001$.

reagennt2 was replaced with reagent2 and incubated for 24 h at room temperature in the dark. Stained slices were stocked in reagent2 and images were taken under Nikon confocal microscope mounted in reagent2.

## Imaging and 3D reconstruction

After slice cleaning and immunostaining, the slices were carefully mounted on glass slides suing R2 reagent. To prevent deformation of brain slices, small agarose pieces matching the thickness of the slices were positioned at the four corners of the slides before placing the coverslip. Imaging was carried out using a Nikon A1R confocal microscope equipped with a 25× objective lens. To minimize Z-axis stretching, images were captured with a fine z-step of 0.5 μm. Subsequently, the captured images underwent denoising via the deep learning method noise2void to enhance the visibility of astrocyte processes (Krull et al, 2018). Following this denoising process, 3D reconstruction was executed using Imaris software. Surfaces were created to build the 3D structure of the astrocytes, and the territory volume was analyzed. These surfaces were then employed to mask the fluorescence signal of astrocytes for further filament analysis.

## Single-cell dissociation

The dissected neocortices were cut into pieces and digested using the Neural Tissue Dissociation Kit (130-092-628, Milteny Biotec) following the manufacturer's instructions. Briefly, the neocortices from pups of postnatal day 7 or 8 were dissected under a microscope and transferred into the ice-cold digestion buffer (with papain and DNase) into one type-C tube. After that, type-C tubes were attached to the gentleMACS Octo Dissociator with Heaters with the gentleMACS Program for 22 min. After digestion, type-C tubes were detached from the Dissociator and washed with 2 ml of 0.04% BSA in 1x HBSS with $Ca^{2+}$, and $Mg^{2+}$. The cell clumps were gently dissociated into single-cell suspension with 1 ml tips. Cells were filtered through a 40 μm filter into a 15 ml tube. They were then centrifuged and washed with 0.04% BSA in 1x HBSS with $Ca^{2+}$, and $Mg^{2+}$ twice to remove myelinated debris. Resuspend cells from each tube in 10% FBS in DMEM without phenol red. GFP-positive cells were collected via fluorescence-activated cell sorting (FACS). Cell viability was qualified by trypan blue staining. Each 10x scRNA library was prepared by pooling FACS sorted cells from 4 pups for each genotype.

## Library preparation for single-cell RNA sequencing

Single-cell capture and library preparation were performed using the 10x Genomics Chromium Single Cell 3' v2 or v3 assays. Single-

cell suspensions were centrifuged at $400 \times g$ for 5 min at 4 °C to gather cells at the bottom of the tube. Remove the supernatant without touching the cell pallet and scale the cell concentration to about 300 cells per μl. 20,000 cells were loaded with the 10x Genomics Chromium Controller for single-cell capture with an efficiency of 50%. The cDNA was amplified for 15 cycles. Then the libraries were constructed according to manufacturer's constructions. The generated paired-end libraries were sequenced on the Illumina Novaseq 6000 platform. After sequencing, raw reads were processed and aligned using the 10x Genomics software Cell Ranger version 5.0.0.

## Single-cell RNA-seq data processing

The expression matrixes of control and Tcf4 cKO group were imported into Seurat and two Seurat objects were created. For each Seurat object, cells with genes detected lower than 200 or higher than 7500 or with a higher percentage of mitochondrial genes than 20 percent were discarded (considered to be low-quality cells that exhibit extensive mitochondrial contamination). The gene counts of each object were normalized. After normalization, two Seurat objects were integrated into one Seurat object according to the tutorial at https://satijalab.org/seurat/archive/v3.2/immune_alignment. To remove the effect of cell cycle genes on cell cluster identification, cell cycle genes were regressed out. The dimension reduction was performed by "RunPCA", followed by RunUMAP to generate a UMAP, and cell clusters were identified by running "FindNeighbors", and "FindClusters(…, resolution = 1)". Cell markers were found by running "FindAllMarkers" and cell clusters were annotated with genes with top rank of gene expression and known genes that are exclusively expressed in different cell types, such as Aldh1l1 for astrocytes or astrocyte precursors, PDGFRα for oligodendrocyte precursors. Umap plots were plotted with "Dimplot" in Seurat. After cluster annotation, clusters with the same identity were combined into one cell type, and the percentage of different cell types was calculated.

To infer developmental trajectory, the Monocle3 pipeline was performed according to the tutorial at https://cole-trapnell-lab.github.io/monocle3/. Based on the results of monocle3, there is a lineage tree started from TIP to APC. To simply reconstruct the trajectory from TIP to APC, we subset TIP and APC clusters and performed trajectory analysis using monocle (Qiu et al, 2017b). Since it have been reported that monocle3 can report lineage that don't exist biologically, to further confirm the lineage diversification after loss of Tcf4, we also performed trajectory analysis using slingshot on the whole dataset (…, reducedDim = 'PCA', clusterLabels = seurat_clusters, start.clus = 0, approx_points = 100) (Street et al, 2018) were used. To identify dynamic gene

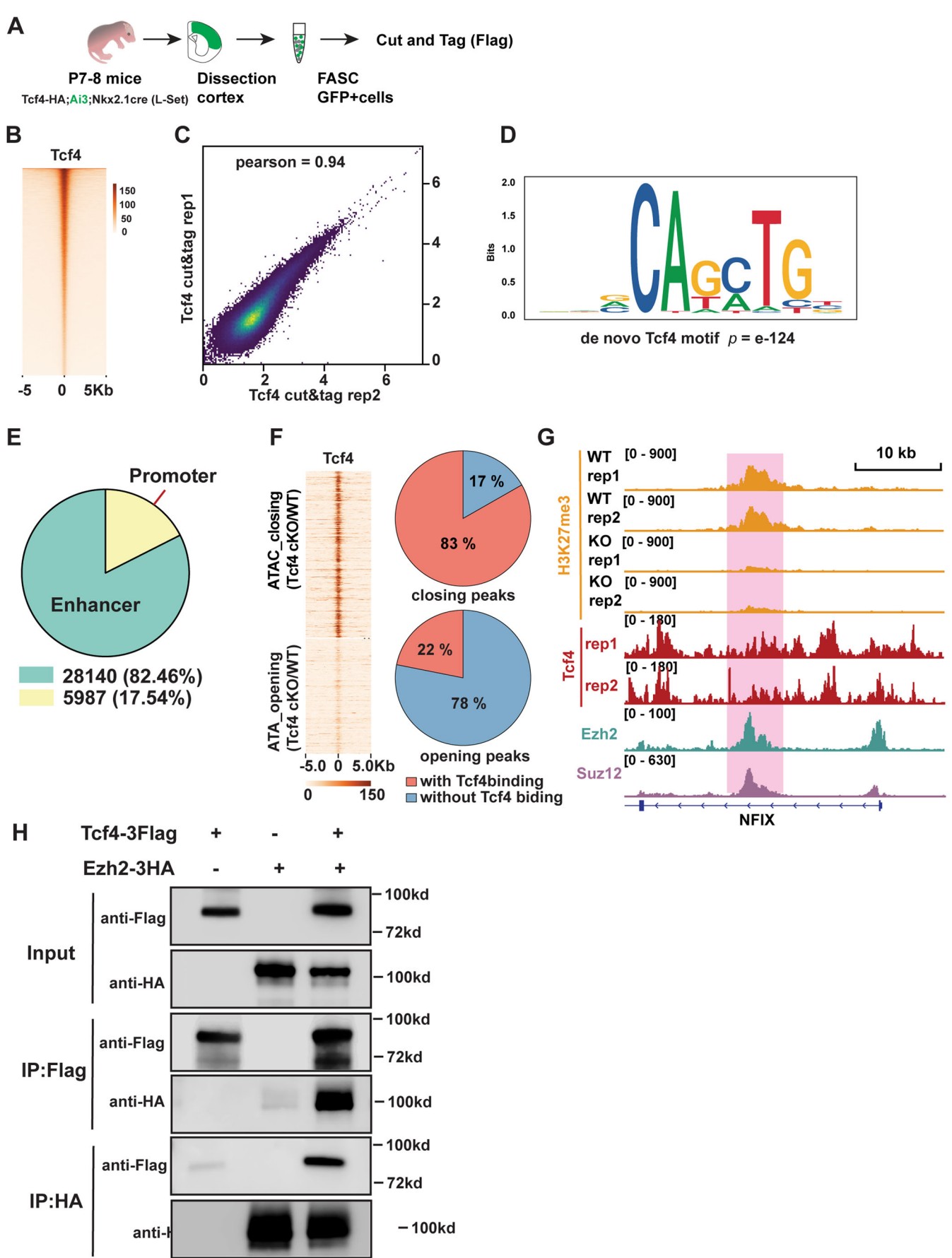

**Figure 7. Tcf4 orchestrates chromatin accessibility to control glial cell fate.**

(A) A schematic diagram illustrates the sample collection and the Cut&Tag workflow for analyzing Tcf4 binding in the genome of cells derived from the Nkx2.1 lineage in the WT neocortices. (B) A heatmap shows the enrichment of Tcf4-binding signals around 5Kb of peak centers. (C) A Pearson correlation highlights the reproducibility of Tcf4 binding across experimental replicates. (D) De novo motif searching by Homer enriched canonical E-box motif of Tcf4. (E) Quantification of types of Tcf4-binding peaks annotated by overlapping with H3K4me1 signals, Encode-predicted enhancers and Ensemble-prebuild regulatory features. (F) The Tcf4-binding activity in differentially accessible regions. Heatmaps show the enrichment of Tcf4-binding signal over closing and opening regions. Pie charts show the percentage of Tcf4-binding peaks overlap with differentially accessible peaks. (G) Track snapshot of Track of CUT and TAG for H3K27me3 and Tcf4, and PRC2 components (Ezh2, Suz12) binding peaks around enhancers of NFIX. The light pink box indicates predicted enhancers of embryonic mouse brains on ENCODE. (H) Co-immunoprecipitation showing Tcf4 interacts with Ezh2.

expression along pseudotime, tradeSeq was used (Van den Berge et al, 2020). After filtering out genes with low levels of pseudotemporal change, cells were clustered into 18 bins with equal pseudotime breaks, and the mean of normalized expression was calculated. The normalized expression of genes was plotted and grouped by hierarchical clustering into 10 initial modules using the R package "pheatmap". The gene ontology was performed using the R package "cluterProfiler" (Yu et al, 2012).

### CUT&Tag library preparation and sequencing

CUT&Tag was carried out as previously described (Kaya-Okur et al, 2020), with minor changes, using Hyperactive In-Situ ChIP Library Prep Kit for Illumina (pG-Tn5) (TD901, Vazyme). Briefly, 50,000 GFP+ cells sorted by FACS were washed and bound to Concanavalin A-coated magnetic beads in Wash buffer (10x Wash buffer, 1X protease inhibitors (A32963, Pierce), rotated for 10 min at RT. Cell-bead slurry was then resuspended in Dig-Wash buffer (Wash buffer containing 0.05% Digtonin) additionally supplemented with 0.1% BSA and 2 mM EDTA, incubated with primary antibody (rabbit anti-H3K4me1/ab8895/Abcam, rabbit anti-H3K27ac/ab4729/Abcam, rabbit anti-H3K27me3/9733s/Cell Signaling Technology) overnight at 4 °C. After incubation with a secondary antibody (Goat anti-Rabbit IgG H&L, ab6702, Abcam) rotated for 1 h at RT, cell-bead slurry was washed 3 times with Dig-Wash buffer and further incubated with pG-Tn5 prepared in Dig-300 buffer (10x Dig-300 buffer, 0.01% Digitonin, 1x protease inhibitors) for 1 h at RT. The cell-bead slurry was then washed 3 times with Dig-300 buffer and subjected to tagmentation in Dig-300 buffer with 10 Mm MgCl₂ for 1 h at 37 °C. To stop tagmentation, the mixture was adjusted to 16.5 mM EDTA and 0.1% SDS and digested with Proteinase K overnight at 37 °C. Tagmented DNA was purified via Phenol-Chloroform extraction with a phase-lock tube (MaXtract High Density Tubes#129046, Qiagen). Further, the libraries were prepared with TruePrep Index Kit V2 for Illumina (TD202, Vazyme) and were purified with DNA Clean Beads (N411, Vazyme). Library quality and quantity were assessed by Thermo Qubit and LabChip, and paired-end sequencing reads (150 bp) were generated on Illumina NovaSeq4000.

For Tcf4 CUT&Tag, sorted cells were centrifuged, and then gently resuspended in Nuclear extraction buffer (20 mM HEPES (PH 7.9), 10 mM KCl, 0.5 mM spermidine, 0.1% Triton X-100, 20% glycerol and 1X protease inhibitors) and incubated on ice for 10 min. After incubation, nuclei were spun down, resuspended in Wash buffer and counted. 2 × 10⁵ Nuclei were subjected to CUT&Tag essentially as described above. Mouse anti-flag antibodies (SLAB0101, Smart-Lifesciences) and Goat anti-Mouse IgG H&L (ab6708, Abcam) were used as primary and secondary antibodies.

### ATAC-seq library preparation and sequencing

ATAC-seq was performed as previously described (Corces et al, 2017), using TruePrep DNA Library Prep Kit V2 for Illumina (TD501, Vazyme). Briefly, 50,000 FACS-sorted GFP+ cells were collected by centrifugation (500 × g, 5 min, 4 °C), and permeabilized for 10 min on ice in 100 µl Nuclear Lysis buffer (320 mM Sucrose, 5 mM CaCl₂, 3 mM Mg(CH₃COO)₂, 0.1 mM EDTA, 10 mM Tris-HCl (PH 8.0), 0.1% NP-40). After centrifugation for 5 min at 2000 × g at 4 °C, pellets were resuspended in 50 µl TruePrep tagment mix (10 µl 5x TTBL, 5 µl TTE Mix, 35 µl Nuclease-free water) and then transposed at 37 °C for 30 min in a PCR amplifier. Tagmented DNA was purified using the MinElute Gel Extraction kit (Qiagen, #28606) and eluted in 26 µl Nuclease-free water. 24 µl DNA were amplified in TruePrep amplify Mix (10 µl 5xTAB, 1 µl TAE, 5 µl PPM) with additional 5 µl barcoded i5/i7 primers (TD202, Vazyme) for 9 cycles according to manufacture instructions. The amplified libraries were purified using SPRIselect beads (Beckman Coulter, Cat# B23319). Library quality and tagmentation quantity were assessed by Thermo Qubit and LabChip, and paired-end sequencing reads (150 bp) were generated on Illumina NovaSeq4000.

### CUT &Tag and ATAC-seq data processing

**Sequence alignment and peak calling**. Paired-end reads were quality- and adaptor-trimmed by TrimGalore (https://www.bioinformatics.babraham.ac.uk/projects/trim_galore/) using default parameters before being aligned to the mm10 reference genome by Bowtie 2 (v2.5.5) (Langmead and Salzberg, 2012). Only aligned pairs were retained to generate the BAM file using Samtools (v1.9) (Li et al, 2009) view with options '-Sbq 1'. Duplicated reads were then removed by Samtools rmdup and only uniquely mapped reads were sorted and indexed for peak calling and visualization. Insert sizes were obtained from the respective BAM files using the Samtool view BAM function and unix commands. All samples displayed the expected periodicity of DNA winding around nucleosomes in genome DNA regions. Highly-correlated replicate BAM files were merged by Samtools merge.

ChIP-seq data of NFIB and NFIX from GSE146793 (Data ref: Fraser et al, 2020), Ezh2 ChIP-seq data from GSE213416 (Data ref: Liu et al, 2023), and Suz12 ChIP-seq data from GSE130628 (Data ref: Wang et al, 2020a) were reanalyzed using above pipeline.

For visualization in Integrative Genomics Viewer (IGV) genome browser, bigwig files were produced by deeptools (v3.4.0) "bamCoverage" with options '-binSize 10 -normalizeUsing RPKM -ignoreForNormalization chrX chrM'.

For CUT&Tag, narrow peaks were called by MACS2 (v2.1.1) (Zhang et al, 2008) with default settings for Tcf4, H3K4me1, and H3K27ac CUT&Tag, and broad H3K27me3 peaks were called with option '-broad'. In the case of ATAC-seq, Peaks were called on individual replicates by MACS2 with options '-nomodel -shift -100

-extsize 200'. Reproducible peaks identified by IDR with acutodff idr ≤0.05 were used in downstream analysis.

### Peak classification

For peak classification the criteria used were as follows: promoter: $-1\,kb$ to $+1\,kb$ around transcription start site (TSS), Intergenic region: downstream region of gene body, and distal region between genes. Exon includes exons and 5' and 3' untranslated regions.

### ATAC-seq peak annotation and statistical analysis of differential accessibility

Differential binding accessibility was carried out by Diffbind packages (https://github.com/hnthirima/DiffBind) using a negative binomial distribution model implemented in DESeq2 with default parameters. MA plot was plotted using ggplot2 package. High confidence differential accessibility peaks (DA peaks) were defined with threshold FDR < 0.05 and log2(fold-change) > 0.5. Peaks were then annotated by ChIPseeker R packages (Yu et al, 2015). Ontology analysis of DA peaks was performed using Genomic Regions Enrichment of Annotation Tool (GREAT) (McLean et al, 2010) with the whole mouse genome (GRCm38/mm10) as the background. Co-localization of DA peaks with Tcf4 binding sites was determined based on the overlap of at least 30% of ATAC-seq peak lengths using deeptools intersectBed. Average signal enrichment values in DA peaks were obtained by deeptools computerMatrix with normalized bigwig files. Heatmaps and profile plots were generated using deeptools plotHeatmap and plotProfile.

### Footprinting analysis

Transcription factor footprints were analyzed using Transcription factor Occupancy prediction By Investigation of ATAC-seq Signal (TOBIAS) (Bentsen et al, 2020; Sung et al, 2016). Merged ATAC-seq BAM files were used as input for TOBIAS ATACorrect, followed by TOBIAS Footprint. TOBIAS BINDetect was run using the following the JASPAR2018 Core databases.

### DNA motif search

The de novo motif search within Tcf4 binding peaks and DA peaks was performed using findMotifsGenome.pl script from Homer packages (http://homer.ucsd.edu/homer/) with the parameter of '-size -75,75 -S 10 -bits'.

### Acute brain slice preparation for calcium imaging

Brain slices were prepared from 10- to 12-week-old mice with AAV injection for Calcium imaging.

Mice were anesthetized with isoflurane and decapitated, and brains were sliced in ice-cold NMDG-HEPES Recovery Solution containing (mM): 14 NMDG, 0.36 KCl, 0.2 $NaH_2PO_4$, 0.45 $NaHCO_3$, 3 HEPES, 0.46 Sodium pyruvate, 1.5 $MgSO_4$, 0.075 $CaCl_2$, 3.75 Glucose, 0.76 Sodium Ascorbates, saturated with 95% $O_2$ and 5% $CO_2$. Brains were cut into 300 μm and incubated at 34 °C for 60 min and subsequently recovered at ~25 °C in artificial cerebrospinal fluid (ACSF) comprising (mM): 126 NaCl, 2.5 KCl, 1.3 $MgCl_2$, 10 Glucose, 2.4 $CaCl_2$, 1.24 $NaH_2PO_4$, and 26 $NaHCO_3$, saturated with 95% $O_2$ and 5% $CO_2$. Then, the calcium signals were recorded under a confocal microscope.

### Calcium imaging analysis

The analysis of Calcium imaging data closely followed the methodology outlined in a previous report (Chai et al, 2017). The time-lapse stacks were adjusted using ImageJ v2.3.0 and the Template Matching plugin. Fluorescence intensity time traces were extracted from the regions of interest (ROIs) and converted to dF/F values utilizing the GECIquant plugin (Srinivasan et al, 2015). The traces were subsequently analyzed in MiniAnalysis, where spontaneous events were manually marked. Event amplitudes, half-widths and frequencies were measured with MiniAnalysis. Events were identified considering amplitudes that exceeded at least two-fold the baseline noise level in the dF/F trace.

### Co-immunoprecipitation (Co-IP)

To verify the interaction between Tcf4 and Ezh2, transfected HEK293T cells were lysed in IP lysis buffer (20 mM HEPES-KOH PH 7.5, 10% glycerol, 10 mM $MgCl_2$, 150 mM NaCl, 1 mM EDTA, 0.5% NP40, and protease inhibitor (A32965, Thermo Scientific), incubated on ice for 30 min. After centrifuged at 12,000 rpm for 20 min at 4 °C, the supernatant was incubated with equilibrated anti-Flag Magnetic beads (M8823, Sigma) or anti-HA magnetic beads (PHM025, Lablead) on an end-to-end rotator overnight at 4 °C. After incubation, the bead slurry were washed five times in IP lysis buffer, and further mixed with 2xSDS-PAGE loading buffer and boiled at 95 °C for 10 min. Co-immunoprecipitation proteins were analyzed by Western blot using indicated antibodies.

### Statistics

In this paper, each data point represents one individual observation. Data are presented as mean ± SEM. To account for intra-class effect within animal, a linear mixed model fit by REML (restricted maximum likelihood) was employed using R package "lmerTest" (Yu et al, 2022). For comparing two groups, a t-test with Satterthwaite's method was utilized. For multi-group comparison, the ANOVA function was utilized to determine the overall $p$-value, and R package "emmeans" were used for $P$-value adjustment for multiple comparisons. A $p$-value of less than 0.05 was considered significant and denoted as follows: $*p < 0.05$, $**p < 0.01$, $***p < 0.001$.

## Data availability

All sequencing data generated in this study have been deposited in the GEO database under accession number GSE244709. Data presented in the main figures have been archived in the BioImage Archive with the accession number S-BIAD1218. All codes used for data analysis were from customized packages available online, and no new code or model was generated in this study. The codes used can be made available upon reasonable request.

The source data of this paper are collected in the following database record: biostudies:S-SCDT-10_1038-S44318-024-00218-x.

## Peer review information

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

## Acknowledgements

We thank all members of the Xie lab for discussions and the IOBS core facility for providing expertise, and we are grateful to Prof. Lan Ma for her supports. This work was supported by STI2030-Major Projects 2021ZD0202304, the National Natural Science Foundation of China (32271020), the Program of Shanghai Academic Research Leader (22XD1400400), Shanghai Municipal Science and Technology Major Project (No.2018SHZDZX01, 19JC1411003), ZJ Lab and Shanghai Center for Brain Science and Brain-Inspired Technology to Y Xie.

## Author contributions

**Yandong Zhang**: Data curation; Formal analysis; Validation; Visualization; Methodology; Writing—review and editing. **Dan Li**: Data curation; Formal analysis; Methodology; Writing—review and editing. **Yuqun Cai**: Data curation. **Rui Zou**: Data curation. **Yilan Zhang**: Data curation. **Xin Deng**: Data curation. **Yafei Wang**: Resources. **Tianxiang Tang**: Resources. **Yuanyuan Ma**: Resources. **Feizhen Wu**: Resources. **Yunli Xie**: Conceptualization; Supervision; Funding acquisition; Writing—original draft; Writing—review and editing.

Source data underlying figure panels in this paper may have individual authorship assigned. Where available, figure panel/source data authorship is listed in the following database record: biostudies:S-SCDT-10_1038-S44318-024-00218-x.

## Disclosure and competing interests statement

The authors declare no competing interests.

# Expanded View Figures

**Figure EV1.   Astrocyte allocation in dorsal neocortex and ventral telencephalon.**

(**A**) Sparse labeling of Emx1 lineage cells induced at E12.5 reveals no astrocytes from the Emx1 lineage observed in the ventral telencephalon at P21. However, in the dorsal neocortex, astrocytes can be traced from the Emx1 lineage generated at E12.5. Sox9 and GFP double-positive astrocytes are indicated by yellow arrowheads. Scale bar: 50 μm. (**B**) Sparse labeling of Nkx2.1 lineage cells induced at E12.5 shows no astrocytes from Nkx2.1 lineage observed in the dorsal neocortex at P21. However, in the ventral telencephalon, astrocytes can be traced from the Nkx2.1 lineage generated at E12.5. Yellow arrowheads indicate Sox9 and GFP double-positive astrocytes. Scale bar: 50 μm. (**C**) A schematic diagram illustrates the location of astrocytes derived from the Emx1 and Nkx2.1 lineages.

▶

      

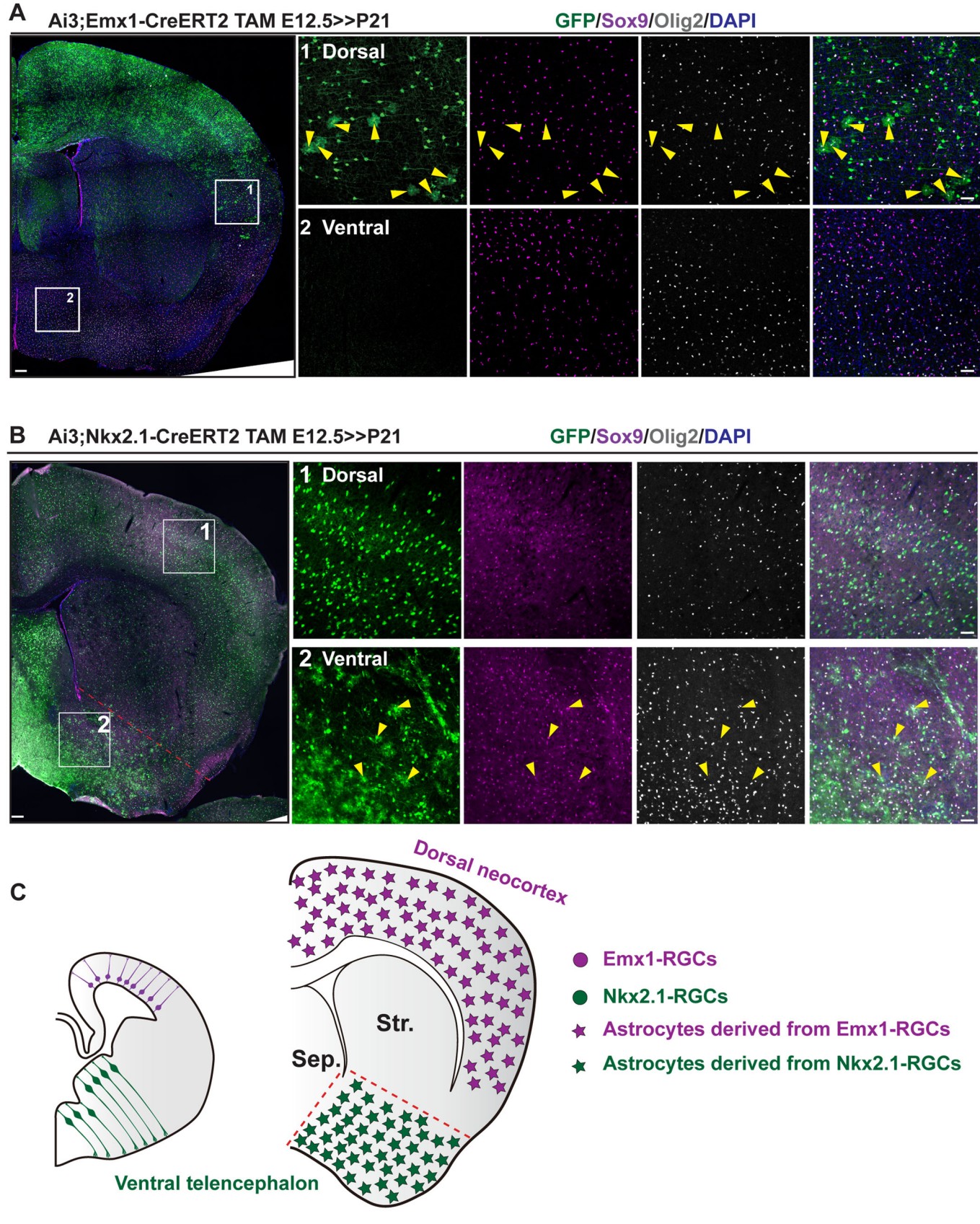

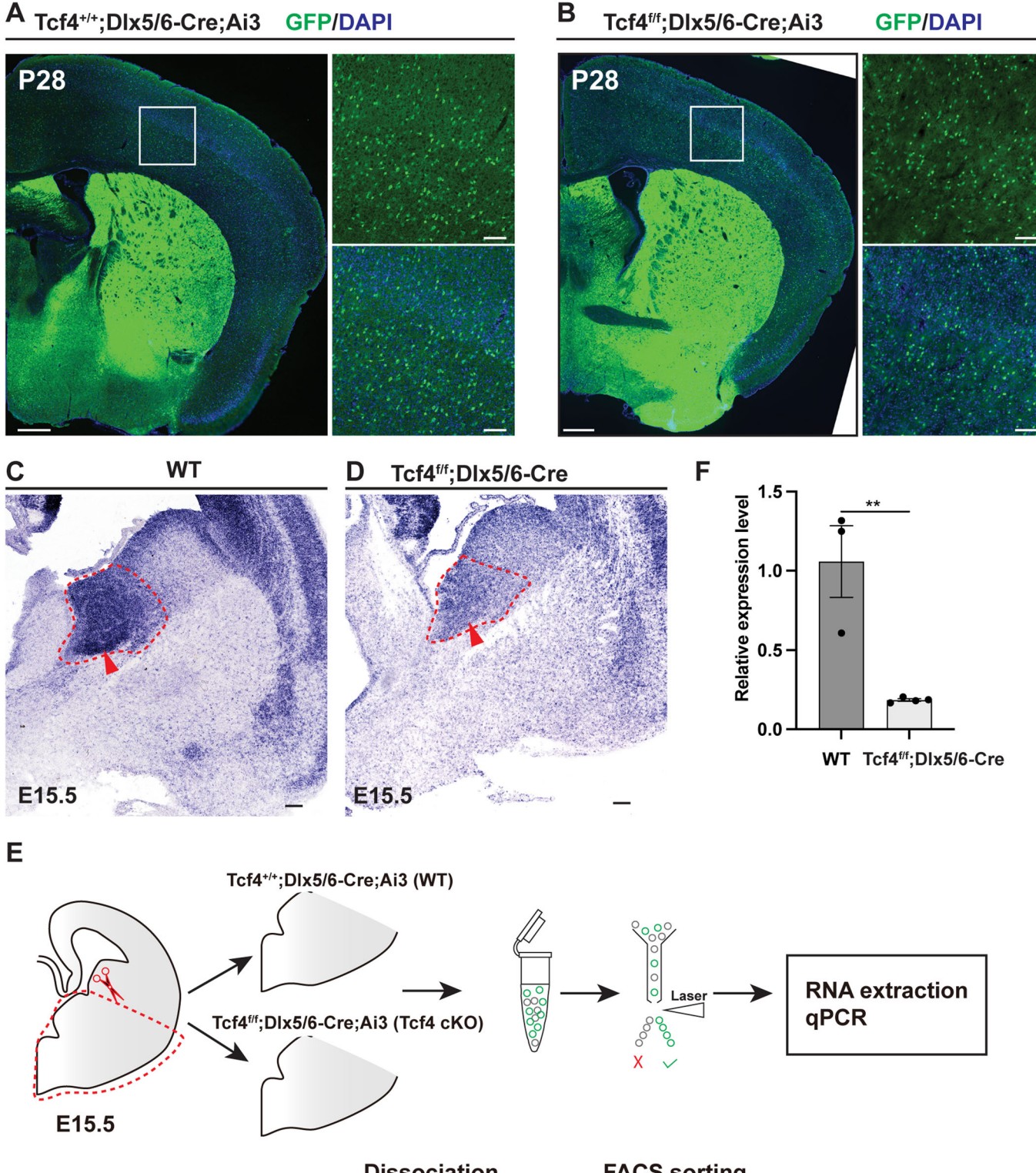

◀ **Figure EV2. Ectopic astrocytes in the dorsal neocortex do not arise from committed progenitors.**

(**A**) Representative images of brain sections from Tcf4$^{+/+}$; Dlx5/6-Cre mice stained for GFP at P28. (**B**) Representative images of brain sections from Tcf4$^{f/f}$; Dlx5/6-Cre mice stained for GFP at P28. Scale bars represent 500 μm and 100 μm respectively. (**C**, **D**) Expression of *Tcf4* was examined by in situ hybridization in the brains of WT and Tcf4$^{f/f}$; Dlx5/6-Cre mice at E15.5. The MGE region is circled with red lines, the red arrowheads indicate the region within which the expression of Tcf4 is decreased, scale bar: 100 μm. (**E**) A schematic diagram illustrates the procedure of collecting the GFP+ cells from WT and Tcf4$^{f/f}$; Dlx5/6-Cre mice and performing real-time quantitative PCR experiments (qPCR). (**F**) The relative expression level of *Tcf4* ($n = 3$ embryos for WT, and $n = 4$ embryos for Tcf4$^{f/f}$; Dlx5/6-Cre). Student's T-test using linear mixed model is used for comparing two groups. $p = 0.0058$. Error bars represent mean ± SEM. **$p < 0.01$.

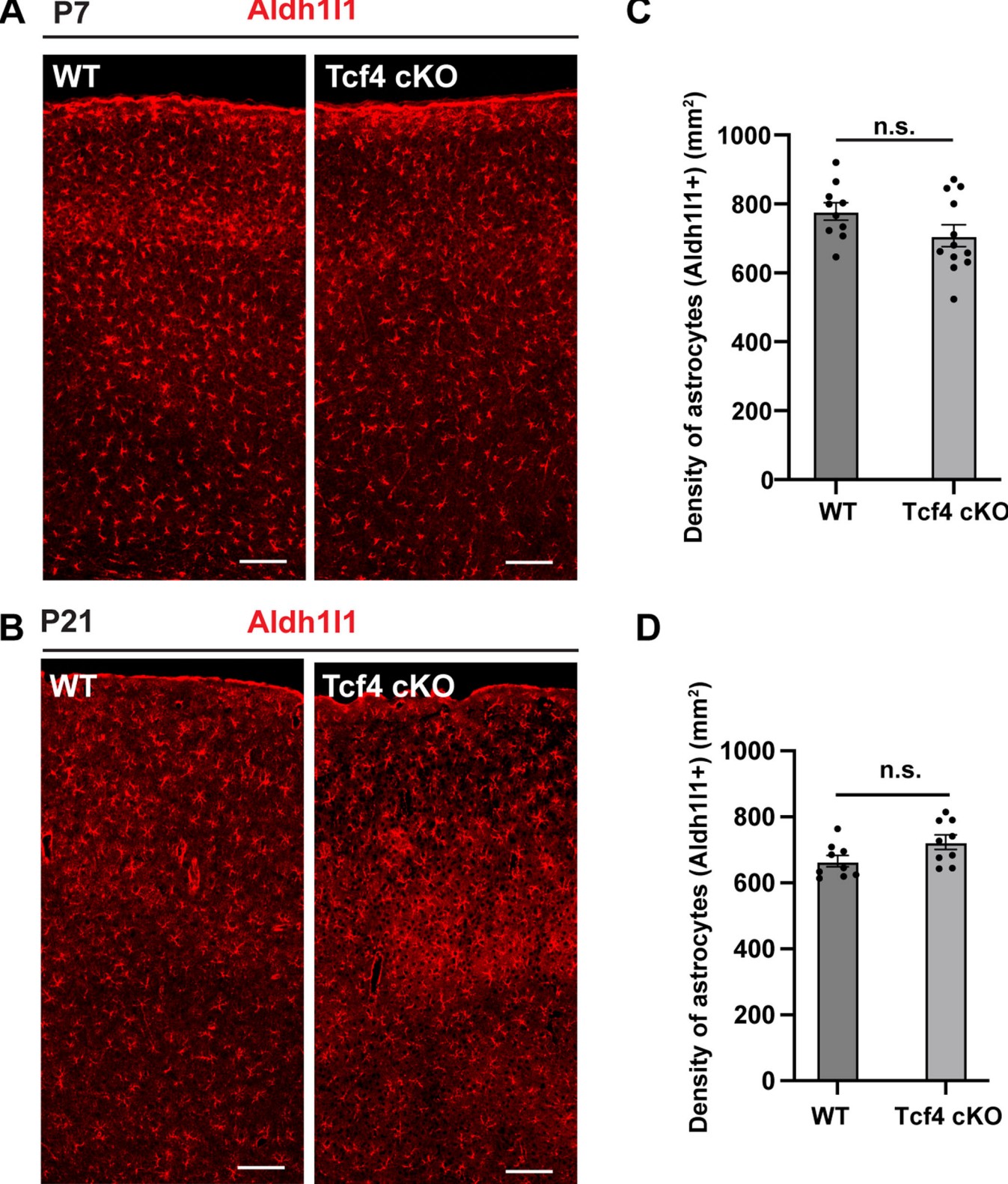

◄ **Figure EV3. The density of Aldh1l1-positive astrocytes in the neocortex was unchanged between Tcf4 cKO and WT brains.**

(A, B) Representative images of brain sections stained for Aldh1l1 in WT and Tcf4 cKO at P7 (**A**) and at P21 (**B**), Scale bar: 100 μm. (**C, D**) Quantification of the density of Aldh1l1+ astrocytes in the neocortex at P7 (**C**), and at P21 (**D**) ($n = 3$ mice were analyzed for each genotype). Comparison between the two groups was conducted using a t-test within a linear mixed model. Error bars represent mean ± SEM.

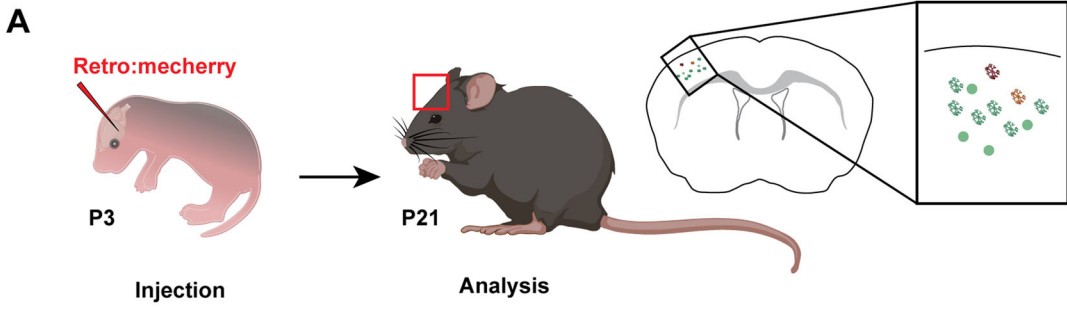

**Figure EV4.  Astrocyte precursor cells generate ectopic astrocytes in the dorsal neocortex of Tcf4 cKO brain.**

(**A**) A schematic diagram illustrates the experimental design. (**B**) Representative images of brain sections in WT and Tcf4 cKO brains stained for GFP, RFP, and Aldh1l1. In WT brains, no RFP-labeled astrocytes co-labeled with GFP were observed, while GFP+ astrocytes in the dorsal neocortex of Tcf4 cKO brains were labeled with RFP. Yellow and white arrowheads indicate astrocytes labeled by retrovirus that are GFP-positive or GFP-negative respectively. Scale bar, 100 μm.

