## [Peer Review File · The EMBO Journal]

Astrocyte allocation during brain development is controlled by Tcf4-mediated fate restriction

Yunli Xie, Yandong Zhang, Dan Li, Yuqun Cai, Rui Zou, Yilan Zhang, Xin Deng, Yafei Wang, Tianxiang Tang, Yuanyuan Ma, and Feizhen Wu

Corresponding author(s): Yunli Xie (yunli.xie@fudan.edu.cn)

Review Timeline:

Submission Date:	26th Jan 24
Editorial Decision:	8th Mar 24
Revision Received:	24th Jun 24
Editorial Decision:	31st Jul 24
Revision Received:	7th Aug 24
Accepted:	9th Aug 24

Editor: Ioannis Papaioannou

Transaction Report:

Dear Yunli,

Thank you for submitting your manuscript EMBOJ-2024-116787 for consideration by The EMBO Journal. It has been seen by three experts in the field, and we have received the full set of their comments, which I have already shared with you (they are included again below). I would also like to thank you for our discussion on the reports and a provisional revision plan, which was very helpful for us to reach a fair and balanced editorial decision.

The referees have provided thorough and detailed reports, and they recognize that the data presented in your manuscript are largely robust and convincing, and the findings novel and interesting. However, they also identify several limitations in the study and the manuscript, and they raise a number of concerns. Importantly, the referees point out that the study does not sufficiently connect the observed cellular effects in the absence of Tcf4 to behavioral or physiological phenotypes in the context of the pathophysiology of neurodevelopmental disorders such as the Pitt-Hopkins syndrome (major concerns 4 and 5 of referee #1, and major points 4 and 5 of referee #2). They suggest, therefore, that the available data are not sufficient to support the claims that the study provides novel insights into the pathophysiology of neurodevelopmental disorders. Furthermore, the referees have a number of additional points regarding gaps and weaknesses in the mechanistic investigation and interpretation of the results, as well as in the presentation of the data in the manuscript.

Given the referees' comments and recommendations, as well as your willingness to embark on a major and substantial revision of your manuscript, I would like to invite you to submit a revised version focusing on a strengthened and well-supported mechanistic investigation of the role of Tcf4 in oligodendrocyte vs. astrocyte specification. Although additional data on the functional consequences of the Tcf4 conditional knock-out with regard to the related neurodevelopmental disorder (such as behavioral analyses) would further strengthen your manuscript and would be desirable in the revised version, they will not be required for further consideration of the manuscript at The EMBO Journal. However, the interpretation and discussion of the findings should be carefully revised throughout the manuscript to accurately reflect the available data, which also applies to the discussion on whether Tcf4 regulates astrocyte allocation. Please also address the comments and suggestions of all three referees in a detailed point-by-point response. I should add that it is EMBO Journal policy to allow only a single round of major experimental revision, and acceptance of your manuscript will therefore depend on the completeness of your responses in this revised version. If you have any questions or comments, we can discuss further in a video call, if you like.

We generally allow three months as standard revision time (June 8, 2024). As a matter of policy, competing manuscripts published during this period will not negatively impact our assessment of the conceptual advance presented by your study. However, we request that you contact us as soon as possible upon publication of any related work, to discuss how to proceed. Should you foresee a problem in meeting this three-month deadline, please let us know in advance and we may be able to grant an extension.

Thank you for the opportunity to consider your work for publication in The EMBO Journal. I look forward to your revision.

Best regards,

Ioannis

Instructions for preparing your revised manuscript

1. When you are ready to submit the revision, please upload:

- A Word file of the manuscript text (including legends of main Figures, EV Figures and Tables). Please make sure that changes are highlighted (or "tracked") to be clearly visible.

- Individual production-quality figure files (one file per figure). When assembling your figures, please refer to our figure preparation guidelines in order to ensure proper formatting and readability in print as well as on screen:

If the data shown in a figure are obtained from n {less than or equal to} 2, please use scatter plots showing the individual data points.

- i. the name of the statistical test used to generate error bars and P values
- ii. the number (n) of independent experiments (please specify technical or biological replicates) underlying each data point (discussion of statistical methodology can be reported in the Materials and Methods section, but figure legends should contain a basic description of n, P, and the test applied)
- iii. the nature of the bars and error bars (s.d., s.e.m.).

- A point-by-point response to the referees' comments, with a detailed description of the changes made (as a word file). All referees' concerns must be fully addressed and their suggestions taken on board. When preparing your letter of response to the referees' comments, please bear in mind that this will form part of the Review Process File and will therefore be available online to the community. Please note that you have the possibility to opt out of the transparent process at any stage prior to publication by letting the editorial office know (contact@embojournal.org); if you do opt out, the Review Process File link will point to the following statement: "No Review Process File is available with this article, as the authors have chosen not to make the review process public in this case.". For more details on our Transparent Editorial Process, please visit our website: <https://www.embopress.org/page/journal/14602075/authorguide#transparentprocess>

- Expanded View (EV) files (replacing Supplementary Information) that are collapsible/expandable online. A maximum of 5 EV Figures can be typeset. EV Figures should be cited as "Figure EV1, Figure EV2" etc. in the text, and their respective legends should be included in the manuscript file after the legends of regular figures. See detailed instructions regarding Expanded View files here:

- For the figures that you do NOT wish to display as Expanded View figures, they should be bundled together with their legends in a single PDF file called "Appendix", which should start with a short Table of Contents (including page numbers). Appendix figures should be referred to in the main text as: "Appendix Figure S1, Appendix Figure S2" etc. Please see detailed instructions here: <https://www.embopress.org/page/journal/14602075/authorguide#expandedview>

- A complete author checklist, which you can download from our author guidelines (<https://www.embopress.org/page/journal/14602075/authorguide>). Please note that the checklist will also be part of the Review Process File.

2. Please note that no statistics should be calculated and shown in Figures if n=2.

3. Before submitting your revision, primary datasets (and computer code, where appropriate) produced in this study need to be deposited in appropriate public databases (see <https://www.embopress.org/page/journal/14602075/authorguide#dataavailability>).

The accession numbers and database should be listed in a formal "Data availability" section (placed after Materials and Methods) that follows the model below (see also

<https://www.embopress.org/page/journal/14602075/authorguide#dataavailability>):

Data availability

- RNA-seq data: Gene Expression Omnibus GSE46843 (<https://www.ncbi.nlm.nih.gov/geo/query/acc.cgi?acc=GSE46843>)
- [data type]: [name of the resource] [accession number/identifier/doi] ([URL or identifiers.org/DATABASE:ACCESSION])

*** All links should resolve to a page where the data can be accessed. ***

*** Please remember to provide in the Data availability section of your revised manuscript reviewer passwords if the datasets are not yet public. ***

*** The Data Availability Section is restricted to new primary data that are part of this study. In case you have no data that require deposition in a public database, please state so instead of referring to the database: "Our study includes no data deposited in public repositories." under the heading "Data availability". ***

4. Please check that the title and the abstract of the manuscript are brief, yet explicit, even to non-specialists. The length of the title should not exceed 100 characters, and the abstract should be a single paragraph not exceeding 175 words.

5. Please also note our reference format: <https://www.embopress.org/page/journal/14602075/authorguide#referencesformat>.

6. At EMBO Press we ask authors to provide source data for the main manuscript figures. Our source data coordinator will contact you to discuss which figure panels we would need source data for and will also provide you with helpful tips on how to upload and organize the files.
7. Please remember: digital image enhancement is acceptable practice, as long as it accurately represents the original data and conforms to community standards. If a figure has been subjected to significant electronic manipulation, this must be noted in the figure legend or in the "Materials and Methods" section. The editors reserve the right to request original versions of figures and the original images that were used to assemble the figure.
8. Our journal encourages inclusion of data citations in the reference list to directly cite datasets that were obtained from public databases. Data citations in the article text are distinct from normal bibliographical citations and should directly link to the database records from which the data can be accessed. In the main text, data citations are formatted as follows: "Data ref: Smith et al, 2001" or "Data ref: NCBI Sequence Read Archive PRJNA342805, 2017". In the Reference list, data citations must be labeled with "[DATASET]". A data reference must provide the database name, accession number/identifiers, and a resolvable link to the landing page from which the data can be accessed at the end of the reference. Further instructions are available at: <https://www.embopress.org/page/journal/14602075/authorguide#referencesformat>.
9. We request authors to consider both actual and perceived competing interests. Please review our policy (<https://www.embopress.org/page/journal/14602075/authorguide#conflictofinterest>) and update your competing interests statement if necessary. Please name this section 'Disclosure and competing interests statement' and place it after the Acknowledgements section.
10. Please note that all corresponding authors are required to provide an ORCID ID upon submission of a revised manuscript (<https://orcid.org/>). Please find instructions on how to link your ORCID ID to your account in our manuscript tracking system in our Author guidelines (<https://www.embopress.org/page/journal/14602075/authorguide#authorshipguidelines>).
11. We use CRediT to specify the contributions of each author in the journal submission system. CRediT replaces the author contribution section, which should be removed from the manuscript. Please use the free text box to provide more detailed descriptions. See also guide to authors: <https://www.embopress.org/page/journal/14602075/authorguide#authorshipguidelines>.
12. Further information is available in our Guide For Authors: <https://www.embopress.org/page/journal/14602075/authorguide>
13. We would also welcome the submission of cover suggestions or motifs to be used by our Graphics Illustrator in designing a cover.
14. Please use the link below to submit your revision:
<https://emboj.msubmit.net/cgi-bin/main.plex>

Referee #1:

In this study, the authors focus on discovering the mechanisms governing astrocyte allocation in distinct brain region. They highlighted a previously unappreciated function of Tcf4 in regulating in astrocyte allocation. Here they found that loss of Tcf4 in ventral Nkx2-1 lineages would lead to alteration of OPC fate into astrocyte precursor in dorsal cortex. Using scRNA-seq, ATAC-seq and Cut&Tag, the authors also define changes in transcriptome and alterations in chromatin accessibility during glia fate transition. The conclusion is quite interesting. However, results of immunostaining, scRNA-seq, ATAC-seq and Cut-tag need to be in-depth and detailed analyzed. Several critical concerns fundamentally undermine the validity, novelty, and interpretative strength of the findings. Below are major and minor comments that detail the substantial issues encountered in this study.

Major Concerns

- 1) Lack of Spatial Analysis for Mislocated Astrocytes: The manuscript presents an interesting premise regarding TCF4's role in astrocyte allocation. However, it significantly lacks detailed spatial analysis of mislocated astrocytes across cortical layers. This omission raises questions about the depth of understanding of TCF4's role and its implications for astrocyte distribution in neurodevelopmental disorders.
- 2) Insufficient Analysis of Subcortical Glia Cell Fate Changes: The study mentions increases in mislocated astrocytes but fails to comprehensively investigate or discuss the fate of glial cells in subcortical regions following TCF4 knockout. This gap in analysis is a critical oversight, given the complex interplay between various glial cell types in brain development and function.

3) Unaddressed Potential Sources of Ectopic Astrocytes: The manuscript does not adequately explore the potential that ectopic astrocytes could arise from neuronal-committed progenitors. This lack of investigation into alternative sources significantly weakens the argument for TCF4's specificity in regulating astrocyte allocation.

4) Overlooked Impact on Oligodendrocytes and Neocortical Network: The effect of TCF4 knockout on oligodendrocyte populations and the overall neocortical network is not addressed. This omission is particularly concerning, given the crucial role of oligodendrocytes in neural function and the potential implications for neurodevelopmental disorders.

5) Absence of Behavioral or Physiological Analysis in Tcf4 cKO Mice: The study fails to connect the observed cellular changes to behavioral or physiological outcomes, particularly in the context of Pitt-Hopkins syndrome. This lack of translational insight significantly limits the study's relevance and applicability to understanding the pathophysiology of neurodevelopmental disorders.

6) Incomplete Time Course Analysis of Astrocyte Density: The manuscript does not provide a comprehensive time course analysis of astrocyte density in different genotypes, nor does it explore the potential modulatory effects of exogenous Tcf4. This lack of data severely hampers the ability to understand the dynamics of astrocyte allocation and the mechanistic role of TCF4.

Minor Issues

1. Figure Presentation and Clarity: Several figures lack essential details such as scale bars and genotype labels, undermining the clarity and interpretability of the presented data. This issue reflects a broader concern regarding the manuscript's overall attention to detail and precision in data presentation. There is lack of scale bar in figure 1L, figure 6I, figure EV1A and 1B, figure EV2 and figure EV7E. The genotype of each panel was missing (EMX1-CreERT2 or NKX2.1-CreERT2) in figure EV1A and EV1B.

2. Inadequate Cell Density Analysis: The analysis of cell density changes in critical figures is superficial and lacks the depth required for a robust interpretation of the data. This shortfall further contributes to the manuscript's inability to convincingly support its conclusions. In Fig 4B,4D and 4F, how does the cell density change?

Referee #2:

EMBOJ-2024-116787-

Comments to authors;

Zhang et al. demonstrated that loss of the transcription factor 4 (Tcf4) in ventral telencephalic neural progenitors alters the fate of oligodendrocyte precursors to transient intermediate astrocyte precursors, resulting in mislocated astrocytes in the dorsal neocortex. Their data are solid based on different comprehensive techniques. However, the mechanism how Tcf4 specifically suppress the transition of OPC to ASP is still not clear. Furthermore, Discussion is not written properly, and should be widely rewritten. Each comment is below.

Major points

1. Backgrounds

Authors should state why they started this study (analysis of Tcf4 cKO mice) to understand the regional allocation of astrocytes in Introduction. Initial association of Tcf4 with regional allocation of astrocytes is not clear.

2. Expression of Tcf4

Authors should show the spatiotemporal distribution of Tcf4 in the ventral telencephalon in the developing stage.

3. Function of Tcf4

It is known that Tcf4 have diverse functions in the developing brain. How can authors confirm that alterations of chromatin accessibility is the principal mechanism for Tcf4-dependent alteration of astrocyte allocation?

4. Are there any behavioral phenotypes in Tcf4 cKO mice? This is important when you consider association of Tcf4 with disease conditions such as Pitt-Hopkins syndrome

5. Are there any phenotypes in heterozygotes of Tcf4 cKO mice? This is also important when you consider association of Tcf4 with disease conditions such as Pitt-Hopkins syndrome

6. Methods and Protocols

The approval numbers for animal experiments and for recombinant gene experiments should be provided.

The detailed methods of tamoxifen administration in the embryonic stages should be presented.

7. Discussion

Authors only summarized their findings in Discussion. They should state the novelty of this study, when compared with previous

studies. Furthermore, they should state limitation of this study.

Referee #3:

In this study, Zhang et al., discover an interesting new role of TCF4 in controlling the oligodendroglial vs astroglial lineage specification. They find that genetic inactivation of TCF4 in ventral-derived oligodendrocyte precursor cells (OPCs), which normally generate oligodendrocytes of the dorsal forebrain, results in the generation of ectopic astrocytes from these progenitors in the dorsal forebrain. They subsequently present a series of elegant mechanistic analyses demonstrating that TCF4 promotes the maintenance of transcriptional and epigenetic programmes controlling oligodendrocyte differentiation, while repressing astrocyte differentiation programmes. Given the important functions of astrocytes and oligodendrocytes, and the poor understanding of the mechanisms underlying their development and diversity, this study provides relevant and novel insights for the fields of fundamental developmental neurobiology and glial biology.

Overall, the study is very well designed and shows apparently robust, convincing findings. However, there are a few technical and conceptual issues that should be addressed.

1) In my view, the title and narrative of the manuscript are a bit misleading, as the study is less focussing on the mechanisms controlling astrocyte allocation, but rather on the mechanisms of astrocyte vs oligodendrocyte fate specification. With the evidence provide, Zhang et al rather show that TCF4 safeguards OPC lineage fate and restricts their specification to an alternative aberrant astrocyte fate, rather than directing the allocation of astrocytes derived from normal astrocyte progenitors. The manuscript should be adapted accordingly.

2) For the single cell and bulk sequencing experiments (Fig 3/5/6/7), the numbers of replicates, QC measures and the variability between replicates are not reported. This makes it currently impossible to judge the quality/reproducibility of the presented data. For example, for the scRNA-seq analysis, UMAP plots showing the distribution of individual samples should be shown as supplementary figure, to allow identifying potential batch effects/inter-sample variability. Figure 1E should be provided with a statistical analysis to support the claim of differential abundance. Similarly, for the ATAC and CUT&RUN experiments, showing multiple replicate coverage plots would allow to get a better understanding of the reproducibility of the findings.

3) In TCF4 cKO, 30% of dorsal astrocytes are from ventral origin. Is there a change in overall astrocyte number/density, or is there a mechanism that reduces generation of dorsal astrocytes to re-balance the overall astrocyte-to-neuron ratio?

4) Although the ectopic astrocytes are probably aberrantly generated from OPCs, it is interesting to see how similar they appear to the locally generated astrocytes, with regard to their functional plasticity and the question to what extent astrocyte properties are dictated by their origin as opposed to their local environment. Vice versa, how similar are the ectopic dorsal astrocytes compared to the ventral astrocytes derived from the same progenitors, that are residing in a different environment/location? Also, it would be interesting to know, although maybe beyond the scope of this study: Is functional interaction of the ventrally generated astrocytes with the dorsal neurons normal? E.g. Astrocyte Ca signals upon neuronal activation, dorsal neuronal activity in TCF4 cKO?

Minor issues:

- General quantifications and statistics:

o Number of cells per section is not an ideal normalisation (e.g. Fig1M), as the section size may vary. It would be better normalise to cells per area

o Using sections or cells as independent replicates for statistical analyses is problematic, as they are not really independent if they are coming from the same animals. It would be better to present the means of cells/sections per animal and perform a t-test on these means per animal (with the animals as independent N), or to use a more appropriate statistical method (linear mixed models) to account for the within-animal correlation; see e.g. Yu et al., Neuron, 2022; However, given the large effect sizes in most experiments reported here, this is a minor issue that probably won't change the interpretation of the data.

- Fig1A-C

o clearer labelling within figure would be helpful (e.g. % of dorsal astrocytes Nkx2.1/YFP+ in Tcf4 cKO)

o have labelled astrocytes in WT been quantified?

- Fig1D-K:

o Red/green color combination difficult for colorvision-impaired readers (also relevant for other immunofluorescence images); magenta instead of red would be better; again, clarify whether quantification of Nkx2.1/YFP+ in Tcf4 cKO

- "When examined at P28, WT brains showed an absence of astrocytes in the neocortex, as expected (Fig. 1A)" - clarify these are YFP labelled astrocytes, not astrocytes in general

- Fig3: More distinct color schemes for the different populations would be helpful. Currently difficult to distinguish

- Methods:

o Include software versions

o In utero electroporation more detailed

o Details how was lineage constructed with Monocle?

o ATAC library prep: "3mM Mg(Ace)₂ I assume this means "Ace" means acetate?

o Code availability: although published packages were used for the analyses, providing the specific analysis code would be helpful for others who want to reproduce the analyses or approaches

- In the results section, the explanations of the rationales of the experiments could be more detailed and explicit. While as specialist in the field, I can see the rationale for the experiments, it is often not very clearly stated explicitly, which will make it more difficult to follow for many readers.

Detailed response to reviewers' comments:

We would like to thank the reviewers for their constructive comments, which have significantly helped us improve our manuscript. Overall, we are pleased that the reviewers recognize our study as providing novel insights into glial development.

In response to their comments, we have conducted additional experiments and analysis. Below, we provide a specific response to each point raised by the reviewers:

Referee #1

In this study, the authors focus on discovering the mechanisms governing astrocyte allocation in distinct brain region. They highlighted a previously unappreciated function of Tcf4 in regulating in astrocyte allocation. Here they found that loss of Tcf4 in ventral Nkx2-1 lineages would lead to alteration of OPC fate into astrocyte precursor in dorsal cortex. Using scRNA-seq, ATAC-seq and Cut&Tag, the authors also define changes in transcriptome and alterations in chromatin accessibility during glia fate transition. The conclusion is quite interesting. However, results of immunostaining, scRNA-seq, ATAC-seq and Cut-tag need to be in-depth and detailed analyzed. Several critical concerns fundamentally undermine the validity, novelty, and interpretative strength of the findings. Below are major and minor comments that detail the substantial issues encountered in this study.

Major Concerns

1) Lack of Spatial Analysis for Mislocated Astrocytes: The manuscript presents an interesting premise regarding TCF4's role in astrocyte allocation. However, it significantly lacks detailed spatial analysis of mislocated astrocytes across cortical layers. This omission raises questions about the depth of understanding of TCF4's role and its implications for astrocyte distribution in neurodevelopmental disorders.

Response: Thank you for this constructive suggestion. We have now analyzed the distribution of mislocated astrocytes across cortical layers and found that these astrocytes are distributed throughout all cortical layers of the neocortex, with more astrocytes located in the deep layers than in the upper layers. This data is presented in a new figure (Figure 1D). Additionally, we analyzed the distribution of mislocated astrocytes from the lateral to medial cortex and found that the number of mislocated astrocyte in the lateral cortex is higher than in the medial cortex. This data is presented in a new figure (Figure 1E).

2) Insufficient Analysis of Subcortical Glia Cell Fate Changes: The study mentions increases in mislocated astrocytes but fails to comprehensively investigate or discuss the fate of glial cells in subcortical regions following TCF4 knockout. This gap in analysis is a critical oversight, given the complex interplay between various glial cell types in brain development and function.

Response: We thank you for the constructive comment.

Additionally, we examined the number of astrocytes and oligodendrocytes derived from the Nkx2.1 lineage in the ventral telencephalon. We found the number of astrocytes (Sox9+) and oligodendrocytes (Olig2+) were not changed upon the loss of Tcf4 (presented in Appendix Figure S3). It is possible that oligodendrocytes and astrocytes have the ability to re-balance the overall cell density (Kessar N et al., 2006, Foerster S et al., 2024). However, at the population level, we could not distinguish newly generated astrocytes in mutant ventral telencephalon following the loss of Tcf4. Therefore, the underlying mechanism remains unclear. Future investigation using lineage tracing analysis at the single cell level will be necessary.

As you suggested, a systematic analysis of glial cell fate in subcortical regions would be interesting. However, we believe this is beyond the scope of the current study, which focuses on the allocation of ectopic astrocytes in the neocortex. Indeed, gliogenesis in subcortical regions is less studied, and we are actively exploring this new area. Nevertheless, we think that the mechanism underlying the transition from OPCs to astrocyte precursors is likely conserved in ventral subcortical regions. We discuss this point further in the manuscript.

3) Unaddressed Potential Sources of Ectopic Astrocytes: The manuscript does not adequately explore the potential that ectopic astrocytes could arise from neuronal-committed progenitors. This lack of investigation into alternative sources significantly weakens the argument for TCF4's specificity in regulating astrocyte allocation.

Response: Thank you for your comment. To explore the potential that ectopic astrocytes could arise from neuron-committed progenitors, we used the Dlx5/6Cre line to deplete Tcf4 in neuron-committed progenitors of the ventral telencephalon. We did not observe an obvious increase of astrocyte in the neocortex (presented in Figures EV2A and 2B). To confirm the depletion of Tcf4 in the neuronal lineage, we first performed in situ hybridization and found that the expression level of Tcf4 was significantly reduced in the ganglion eminence (MGE) (presented in Figures EV2C and 2D). Note that the remaining low-level expression of Tcf4 in Tcf4^{fl/fl};Dlx5/6Cre brains is most likely derived from glial lineages, as the expression of Tcf4 was not altered in these cells. Additionally, we enriched the GFP+ cells from WT and Tcf4^{fl/fl};Dlx5/6Cre brains by FACS and examined the level of Tcf4 mRNA by quantitative real-time PCR (q-PCR). Consistently, we found that the expression of Tcf4 was significantly reduced upon Dlx5/6Cre-mediated recombination (presented in Figures EV2E and 2F). Taken together, these results demonstrate that ectopic astrocytes did not arise from neuron-committed progenitors.

4) Overlooked Impact on Oligodendrocytes and Neocortical Network: The effect of TCF4 knockout on oligodendrocyte populations and the overall neocortical network is not addressed. This omission is particularly concerning, given the crucial role of oligodendrocytes in neural function and the potential implications for neurodevelopmental disorders.

Response: Thank you for the constructive comment. To examine the effect of Tcf4 knockout on oligodendrocyte populations, we stained sections from both WT and Tcf4 cKO brains at P28. We observed a slight decrease, though not statistically significant, in olig2+ oligodendrocytes in Tcf4 cKO neocortex compared to the WT neocortex (presented in Appendix Figure S2A-C). Immunostaining for mature oligodendrocytes, marked by CC1, showed no obvious change in their number following the loss of Tcf4 (presented in Appendix Figure S2D-F). This finding is intriguing given previous reports that OPCs can compensate for changes in OPC numbers from different origins (Kessar N et al., 2006, Foerster S et al., 2024). It is possible that the OPC density is rebalanced at the population level in the neocortex during the fate transition from OPCs to APCs in Tcf4 mutants. This could potentially impact neuronal functions. Interestingly, we observed a similar neuronal investment in both local and ectopic astrocytes (Figures 2F and 2G), suggesting that both local and ectopic astrocytes interact with a comparable number of neuronal somata. Therefore, these data suggest that OPCs and ectopic astrocytes adapt to the local environment to fulfill their functions.

5) Absence of Behavioral or Physiological Analysis in Tcf4 cKO Mice: The study fails to connect the observed cellular changes to behavioral or physiological outcomes, particularly in the context of Pitt-Hopkins syndrome. This lack of translational insight significantly limits the study's relevance and applicability to understanding the pathophysiology of neurodevelopmental disorders.

Response: We thank you for the constructive suggestion. It is particularly interesting to examine the behavioral outcomes in Tcf4 cKO mice. After discussing with the editor, we agree that the behavioral tests may extend beyond the scope of the current study. However, we tempered our conclusions related to the physiological aspects of Pitt-Hopkins syndrome. Nevertheless, we appreciate your valuable comments and will address this important question in the future research.

6) Incomplete Time Course Analysis of Astrocyte Density: The manuscript does not provide a comprehensive time course analysis of astrocyte density in different genotypes, nor does it explore the potential modulatory effects of exogenous Tcf4. This lack of data severely hampers the ability to understand the dynamics of astrocyte allocation and the mechanistic role of TCF4.

Response: Thank you for this constructive comment. We first analyzed the density of ectopic astrocytes during cortical development. We observed an increase in ectopic astrocytes from P15 to P21, after which their number remained stable into adulthood (presented in Figures 1N and 1O). Furthermore, to analyze the entire astrocyte population during cortical development, we stained sections from both WT and Tcf4 cKO brains with the pan-astrocyte marker Aldh1l1 at P7, when astrocytes are immature, and at P21, when they are mature. We found that the total number of astrocytes was similar in both WT and Tcf4 cKO neocortex at both early (P7) and later (P21) stages (presented in Figure EV3). These data suggest that the density of astrocytes rebalanced in Tcf4 cKO brains as ectopic astrocytes increased. We have discussed this point in the manuscript.

While exogenous manipulation of Tcf4 through overexpression may provide further insights into astrocyte allocation, it is technically challenging to specifically express Tcf4 in astrocytes in vivo. However, our loss-of-function analysis has elucidated the mechanism by which Tcf4 regulates astrocyte allocation. Based on single-cell RNA-seq, ATAC-seq, and Cut-Tag experiments from the Nkx2.1 lineage sorted from the dorsal neocortex, we found that the loss of Tcf4 led to the activation of genes related to astrogenesis in OPCs of the dorsal neocortex. We propose that Tcf4-mediated transcription represses astrocyte fate in the OPC lineage of the dorsal neocortex.

Minor Issues

1. Figure Presentation and Clarity: Several figures lack essential details such as scale bars and genotype labels, undermining the clarity and interpretability of the presented data. This issue reflects a broader concern regarding the manuscript's overall attention to detail and precision in data presentation. There is lack of scale bar in figure 1L, figure 6I, figure EV1A and 1B, figure EV2 and figure EV7E. The genotype of each panel was missing (EMX1-CreERT2 or NKX2.1-CreERT2) in figure EV1A and EV1B.

Response: We have now carefully gone through the figures and provided detailed information.

2. Inadequate Cell Density Analysis: The analysis of cell density changes in critical figures is superficial and lacks the depth required for a robust interpretation of the data. This shortfall further contributes to the manuscript's inability to convincingly support its conclusions. In Fig 4B,4D and 4F, how does the cell density change?

Response: We thank you for your comments. We understand the importance of a thorough and detailed analysis to support the conclusions. To address this, we have conducted a more comprehensive analysis of cell density changes. We analyzed the spatial changes of astrocyte density and found that ectopic astrocytes were distributed throughout all cortical layers of the neocortex, with more astrocytes located in the deep layers than in the upper layers. In addition, when analyzing the distribution of mislocated astrocytes from the lateral to medial cortex, we found that the number of mislocated astrocyte in the lateral cortex is higher than in the medial cortex (Figures 1A-1E). We think cell density is a more accurate measure than cell number for reflecting the phenotype. Therefore, we re-analyzed the data presented in the original Figures 4B, 4D and 4F. We found that the density of APCs (positive for *Blbp* or *Aldh111*) is increased in the *Nkx2.1* lineage in the dorsal neocortex of *Tcf4* cKO brains compared to the WT brains, while the density of OPCs is reduced in the *Nkx2.1* lineage in the dorsal neocortex of *Tcf4* cKO brains (presented in Figures 4B, 4D and 4F). These findings are consistent with the single-cell RNA-seq data, which show a transition from OPCs to APCs in the *Nkx2.1* lineage in the dorsal neocortex upon the loss of *Tcf4*.

Referee #2

*EMBOJ-2024-116787-
Comments to authors;*

Zhang et al. demonstrated that loss of the transcription factor 4 (Tcf4) in ventral telencephalic neural progenitors alters the fate of oligodendrocyte precursors to

transient intermediate astrocyte precursors, resulting in mislocated astrocytes in the dorsal neocortex. Their data are solid based on different comprehensive techniques. However, the mechanism how Tcf4 specifically suppress the transition of OPC to ASP is still not clear. Furthermore, Discussion is not written properly, and should be widely rewritten. Each comment is below.

Major points

1. Backgrounds

Authors should state why they started this study (analysis of Tcf4 cKO mice) to understand the regional allocation of astrocytes in Introduction. Initial association of Tcf4 with regional allocation of astrocytes is not clear.

Response: Thank you for this comment. Building on our previous work on Tcf4 in the excitatory neuronal lineage (Wang et al., Cereb. Cortex, 2020), we found that Tcf4 is highly expressed in the ventral telencephalon where Nkx2.1-expression progenitors located. We reasoned that Tcf4 might play a role in the regulation of Nkx2.1 progenitors. When we conditionally depleted Tcf4 in the Nkx2.1 lineage, we found that Tcf4 is essential for regional astrocyte allocation. We have included this information in the introduction.

2. Expression of Tcf4

Authors should show the spatiotemporal distribution of Tcf4 in the ventral telencephalon in the developing stage.

Response: Thank you for this advice. Due to the lack of a convincing specific antibody against the Tcf4 protein, we have performed in situ hybridization using Tcf4 probes to examine the spatiotemporal expression of Tcf4 in the ventral telencephalon. We found that Tcf4 is highly expressed in the medial ganglionic eminence (MGE) region during the developing stages (presented in Appendix Figure S1).

3. Function of Tcf4

It is known that Tcf4 have diverse functions in the developing brain. How can authors confirm that alterations of chromatin accessibility is the principal mechanism for Tcf4-dependent alteration of astrocyte allocation?

Response: Thank you for this comment. As a transcription factor, Tcf4 has diverse functions in brain development, primarily through the regulation of gene expression. In our study, we provided an in-depth analysis of how Tcf4 regulates its target genes. We found that chromatin accessibility was altered upon the loss of Tcf4 in the Nkx2.1 lineage, which further influenced target gene expression. Particularly, we found that in the Nkx2.1 lineage, genes regulated by Tcf4 are associated with astrogenesis as revealed by Cut&Tag experiments (Figures 5 and 6). In this study, we demonstrated that the OPC fate

is restricted through Tcf4-mediated repression of astrocytic genes, which resulted in mislocation of astrocytes with ventral origin in the dorsal neocortex.

4. Are there any behavioral phenotypes in Tcf4 cKO mice? This is important when you consider association of Tcf4 with disease conditions such as Pitt-Hopkins syndrome

Response: We thank you for the constructive suggestion. Reviewer 1 also mentioned this direction. It is particularly interesting to examine the behavioral outcomes in Tcf4 cKO mice. After discussing with the editor, we agree that the behavioral tests may extend beyond the scope of the current study. However, we tempered our conclusions related to the physiological aspects of Pitt-Hopkins syndrome. Nevertheless, we appreciate your valuable comments and will address this important question in the future research.

5. Are there any phenotypes in heterozygotes of Tcf4 cKO mice? This is also important when you consider association of Tcf4 with disease conditions such as Pitt-Hopkins syndrome

Response: We have examined the phenotypes in the heterozygotes of Tcf4 cKO mice and did not find an obvious increase in ectopic astrocytes in the dorsal neocortex compared to the WT brains. It is likely that a lower amount of Tcf4 is sufficient to maintain the OPC fate in the heterozygotes. We now tempered our conclusions related to the physiological aspects of Pitt-Hopkins syndrome.

6. Methods and Protocols

The approval numbers for animal experiments and for recombinant gene experiments should be provided.

The detailed methods of tamoxifen administration in the embryonic stages should be presented.

Response: We have now provided the detailed information.

7. Discussion

Authors only summarized their findings in Discussion. They should state the novelty of this study, when compared with previous studies. Furthermore, they should state limitation of this study.

Response: Thank you for the constructive suggestion. We have updated the discussion to highlight the novelty of our work. Additionally, we also discussed the limitations of the study.

Referee #3

In this study, Zhang et al., discover an interesting new role of TCF4 in controlling the oligodendroglial vs astroglial lineage specification. They find that genetic inactivation of TCF4 in ventral-derived oligodendrocyte precursor cells (OPCs), which normally generate oligodendrocytes of the dorsal forebrain, results in the generation of ectopic astrocytes from these progenitors in the dorsal forebrain. They subsequently present a series of elegant mechanistic analyses demonstrating that TCF4 promotes the maintenance of transcriptional and epigenetic programmes controlling oligodendrocyte differentiation, while repressing astrocyte differentiation programmes. Given the important functions of astrocytes and oligodendrocytes, and the poor understanding of the mechanisms underlying their development and diversity, this study provides relevant and novel insights for the fields of fundamental developmental neurobiology and glial biology.

Overall, the study is very well designed and shows apparently robust, convincing findings. However, there are a few technical and conceptual issues that should be addressed.

1) In my view, the title and narrative of the manuscript are a bit misleading, as the study is less focussing on the mechanisms controlling astrocyte allocation, but rather on the mechanisms of astrocyte vs oligodendrocyte fate specification. With the evidence provide, Zhang et al rather show that TCF4 safeguards OPC lineage fate and restricts their specification to an alternative aberrant astrocyte fate, rather than directing the allocation of astrocytes derived from normal astrocyte progenitors. The manuscript should be adapted accordingly.

Response: We propose that the repression of astrocyte lineage development in the ventral NKX2.1 lineage within the dorsal neocortex is a novel mechanism to safeguard regional astrocyte allocation. We provided a detailed mechanism explaining how astrocyte fate in the Nkx2.1 lineage is restricted in the dorsal neocortex. In the WT neocortex, the restricted fate of OPCs originating from the ventral telencephalon ensures allocation of astrocytes within the ventral telencephalon exclusively. Therefore, fate restriction plays a crucial role in the regional allocation of astrocytes. However, If the reviewer insists that we should change the title, we are willing to revise it accordingly.

2) For the single cell and bulk sequencing experiments (Fig 3/5/6/7), the numbers of replicates, QC measures and the variability between replicates are not reported. This makes it currently impossible to judge the quality/reproducibility of the presented data. For example, for the scRNA-seq analysis, UMAP plots showing the distribution of individual samples should be shown as supplementary figure, to allow identifying potential batch effects/inter-sample variability. Figure 1E should be provided with a statistical analysis to support the claim of differential abundance. Similarly, for the ATAC

and CUT&RUN experiments, showing multiple replicate coverage plots would allow to get a better understanding of the reproducibility of the findings.

Response: Thank you for your constructive comments. To enrich cells derived from the Nkx2.1 lineage in the dorsal neocortex and to minimize the bench effects on scRNA-seq, we pooled the sorted cells from at least four brains for each genotype. In the single-cell sequencing dataset of the WT group, the median of unique molecular identifiers (UMI) per cell is 6,296, and the median number of genes detected per cell is 2,997 (presented in Appendix Figure S4A). For the Tcf4 cKO group, the median UMI per cell is 9,027, and the median number of genes detected per cell is 3,680 (presented in Appendix Figure S4B). When analyzing data quality, we found a very low percentage of genes from mitochondrial and red blood cells (presented in Appendix Figure S4A and 4B), suggesting high data quality. We have incorporated this information into the manuscript. In addition, we performed another scRNA-seq for the Nkx2.1 lineage in the dorsal neocortex for a separate project. When integrated with the current dataset, we obtained the same cluster of cell types and observed an increase of TIP cells (See below). Therefore, the data presented in the manuscript is highly reproducible.

For the ATAC-seq and Cut&Tag experiments, we generated dataset from two independent experiments. As suggested, we presented the multiple replicate coverage plots (Figure 6 G and H, Figure 7G). We also calculated Pearson's correlation coefficients between repeats for each genotype in ATAC-seq and Cut&Tag experiments, all of which showed high correlation coefficients, indicating high reproducibility. These data are presented in Appendix Figures S5 and S6.

3) In TCF4 cKO, 30% of dorsal astrocytes are from ventral origin. Is there a change in overall astrocyte number/density, or is there a mechanism that reduces generation of dorsal astrocytes to re-balance the overall astrocyte-to-neuron ratio?

Response: Thank you for the constructive comments. To analyze the effect of ventrally derived astrocytes on local astrocyte population in the neocortex, we quantified astrocyte density in the dorsal neocortex using Aldh1l1 staining. Our analysis showed no significant change in the overall astrocyte density (Figure EV3). Thus, as you suggested, there may be an underlying mechanism that reduces generation of dorsal astrocytes to maintain the overall astrocyte-to-neuron ratio. We plan to investigate this mechanism in detail in the future.

4) Although the ectopic astrocytes are probably aberrantly generated from OPCs, it is interesting to compare them to the locally generated astrocytes, with regard to their functional plasticity and the question to what extent astrocyte properties are dictated by their origin as opposed to their local environment. Vice versa, how similar are the ectopic dorsal astrocytes compared to the ventral astrocytes derived from the same progenitors, that are residing in a different environment/location? Also, it would be interesting to know, although maybe beyond the scope of this study: Is functional interaction of the ventrally generated astrocytes with the dorsal neurons normal? E.g. Astrocyte Ca signals upon neuronal activation, dorsal neuronal activity in TCF4 cKO?

Response: Thank you for these constructive comments. To understand how the local environment influences ectopic astrocytes, we examined the morphological characteristics and calcium activity of both ectopic astrocytes and the local astrocytes. To label the local astrocytes within the dorsal neocortex, we performed in utero electroporation to label Emx1-RGCs of the dorsal neocortex at E16.5 with plasmids expressing GFP. We performed brain clearing using the CUBIC method to obtain high-quality images. These images were subsequently denoised via the deep learning method noise2void to enhance the visibility of astrocyte fine processes. We found that ectopic astrocytes exhibit a similar morphology to local astrocytes within the dorsal neocortex. In the ventral telencephalon, mutant astrocytes show a similar morphology to WT astrocytes, suggesting that Tcf4 depletion did not result morphological effects on astrocytes in the ventral telencephalon. Interestingly, we found that the complexity of astrocytes within the dorsal neocortex is higher than in the ventral telencephalon, suggesting that the existence of the astrocyte plasticity. This finding is consistent with our results showing that ectopic astrocytes exhibit a similar calcium activity to local astrocytes both spontaneous and upon phenylephrine stimulation. These results are presented in Figure 2.

We agree that investigating the plasticity of ectopic astrocytes in the dorsal neocortex is indeed of interesting, particularly regarding the functional interactions of the ventrally generated astrocytes with dorsal neurons upon neuronal activation. However, our current study focuses on the mechanisms underlying the generation of ectopic astrocytes to provide insights into the regional astrocyte allocation. Therefore, a systemic investigation of the plasticity of ectopic astrocytes is beyond the scope of this study as you mentioned. However, we appreciate your valuable comments and will address this important question in the future research.

Minor issues:

- General quantifications and statistics:

o Number of cells per section is not an ideal normalisation (e.g. Fig1M), as the section size may vary. It would be better normalise to cells per area

Response: Thank you for the suggestion. We now presented the data as cell density.

o Using sections or cells as independent replicates for statistical analyses is problematic, as they are not really independent if they are coming from the same animals. It would be better to present the means of cells/sections per animal and perform a t-test on these means per animal (with the animals as independent N), or to use a more appropriate statistical method (linear mixed models) to account for the within-animal correlation; see e.g. Yu et al., Neuron, 2022; However, given the large effect sizes in most experiments reported here, this is a minor issue that probably won't change the interpretation of the data.

Response: Thank your for these constructive suggestions. As you mentioned, we have included a larger number of samples in our experiments for analysis. Indeed, when we used linear mixed models to account for within-animal correlation, the interpretation of the data remained the same. We updated this information accordingly in the manuscript.

- Fig1A-C

*o clearer labelling within figure would be helpful (e.g. % of dorsal astrocytes Nkx2.1/YFP+ in Tcf4 cKO)
o have labelled astrocytes in WT been quantified?*

Response: We have revised the label accordingly as “% of astrocytes (GFP+) in the neocortex of Tcf4 cKO”. Additionally, we provided detailed information in the figure legends accordingly. Since we did not observe any astrocytes derived from the Nkx2.1 lineage in the dorsal neocortex of WT mice (Fig. 1A), we did not include the quantification (the number is 0) of these astrocytes.

- Fig1D-K:

o Red/green color combination difficult for colorvision-impaired readers (also relevant for other immunofluorescence images); magenta instead of red would be better; again, clarify whether quantification of Nkx2.1/YFP+ in Tcf4 cKO

Response: Thank you for this suggestion. We have changed the red color to magenta. In addition, we clarified the quantification as “% of astrocytes (GFP+) in the neocortex of Tcf4 cKO”.

- "When examined at P28, WT brains showed an absence of astrocytes in the neocortex, as expected (Fig. 1A)" - clarify these are YFP labelled astrocytes, not astrocytes in general

Response: We clarified that these are YFP-labelled astrocytes derived from the Nkx2.1 lineage, rather than astrocytes in general.

- Fig3: More distinct color schemes for the different populations would be helpful. Currently difficult to distinguish

Response: To make it easier to distinguish, we used distinct color schemes for the different cell populations.

- Methods:

o Include software versions

o In utero electroporation more detailed

o Details how was lineage constructed with Monocle?

o ATAC library prep: "3mM Mg(Ace)2 I assume this means "Ace" means acetate?

o Code availability: although published packages were used for the analyses, providing the specific analysis code would be helpful for others who want to reproduce the analyses or approaches

Response: Thank you for these constructive suggestions.

- We included the software versions in the method section.
- We provided detailed information for in utero electroporation.
- We provided detailed information for lineage construction with Monocle3.
- We corrected Mg(Ace)2 as Mg(CH₃COO)₂.
- We provided precise information on the published codes used in the study.

- In the results section, the explanations of the rationales of the experiments could be more detailed and explicit. While as specialist in the field, I can see the rationale for the experiments, it is often not very clearly stated explicitly, which will make it more difficult to follow for many readers.

Response: Thank you for the constructive suggestion. In the revised manuscript, we provided explanations of the rationales behind the experiments to make it easier to follow.

Dear Yunli,

Thank you for the submission of your revised manuscript to The EMBO Journal and your patience during its peer review. It has now been seen by the three original referees who previously assessed the earlier version of your manuscript, and we have received the full set of their comments (included below). As you will see, all referees are satisfied with the revision and acknowledge that the work has been substantially improved. Referees #1 and #3 have no further concerns, while referee #2 has a few suggestions for the improvement of the Introduction and the Discussion, as well as for a clarification in the Materials and Methods, which we kindly ask you to address in a final revised version of your manuscript. Please describe in detail all changes to the manuscript in a cover letter or a point-by-point response.

From the editorial side, there are also a few changes/corrections that we need from you before we can proceed with acceptance of your manuscript for publication in The EMBO Journal:

- Please note that the full funding information should be entered in our manuscript handling system during resubmission of your manuscript, and it should be identical to the information provided in the Acknowledgements section of your manuscript. The information related to "ZJ Lab and Shanghai Center for Brain Science and Brain-Inspired Technology" is currently missing from our online system.
- Please provide a list of up to 5 relevant keywords after the Abstract of your revised manuscript.
- Please rename the heading "Data availability and code availability" to "Data availability".
- Please make sure that all deposited datasets are publicly available; the reviewer access codes can now be removed from the Data availability section, while the permanent URLs to the datasets must be provided/updated.
- Please note that the Data availability section is restricted to new datasets that were generated in this study (not previously deposited datasets/datasets from previous studies). Previously generated/reported datasets that were re-used/re-analyzed in this study should be cited in the text and the References list according to our instructions about "Data citations": <https://www.embopress.org/page/journal/14602075/authorguide#referencesformat>.
- Please make sure that all requested Source Data have been included in your BiImage Archive submission, and that they are all clearly annotated (i.e. linked to their respective Figures).
- Please rename the heading of your conflict-of-interest statement to "Disclosure and competing interests statement".
- The author contributions statement should be removed from the manuscript file. Instead, we now use CRediT to specify the contributions of each author in the journal submission system. Please feel free to use the free text box to provide more detailed descriptions during submission. See also our guide to authors for more information: <https://www.embopress.org/page/journal/14602075/authorguide#authorshippinguidelines>.
- We noticed that callout(s) of Fig. 7H is/are missing; please make sure that all Figure panels are called out as appropriate in your revised manuscript.
- Please correct the callout of Appendix Figure 1 to "Appendix Figure S1".
- Please note that EMBO press papers are accompanied online by:
 - A) a short (2 sentences) summary of the findings and their significance,
 - B) 2-5 short bullet points highlighting the key results, and
 - C) a synopsis image in .jpg or .png format that is exactly 550 pixels wide and 300-600 pixels high (the height is variable). Please note that the text needs to be legible at the final size. Please upload this information along with your revised manuscript (the text for A and B should be provided in a separate Word file).
- Please define the annotated p values ** as well as provide the exact p-values for the same in the legend of Figure EV 2f as appropriate.
- Please note that the exact p values should be provided in the legends of Figures 2d; 4b, d, f; 6i.
- Please indicate the statistical test used for data analysis in the legends of Figures 5g-h; 6d.
- Please note that for Figures 1c-e, j, o; 2d-2, g, k; 4b, d, f; p-values and statistical tests are indicated in the legends. However, comparison for the same, "****/**/****" has not been represented in the figures. Please rectify this in the figures or legends as applicable.

- Please note that information related to "n" is missing in the legend of Figure EV 2f.
- Please note that the scale bar needs to be defined in the legends of Figures EV 1a-b; EV 3a-b.
- Please note that the white arrowheads are not defined in the legend of Figure 2h. This needs to be rectified.

Please also note that as part of the EMBO publications' Transparent Editorial Process, The EMBO Journal publishes online a Peer Review File along with each accepted manuscript. This File will be published in conjunction with your paper and will include the referee reports, your point-by-point response and all pertinent correspondence relating to the manuscript. You can opt out of this by letting the editorial office know (contact@embojournal.org). If you do opt out, the Peer Review File link will point to the following statement: "No Peer Review File is available with this article, as the authors have chosen not to make the review process public in this case."

We look forward to seeing a final version of your manuscript as soon as possible. Please use this link to submit your revision: <https://emboj.msubmit.net/cgi-bin/main.plex>

Best regards,

Ioannis

Referee #1:

I have no more question.

Referee #2:

Comments to authors;

The manuscript was partially improved, but some questions I raised have not been responded properly.

Major points

1. Backgrounds

I could not find the parts below which authors suggested in the new version of Introduction.

...Building on our previous work on Tcf4 in the excitatory neuronal lineage (Wang et al., *Cereb. Cortex*, 2020), we found that Tcf4 is highly expressed in the ventral telencephalon where Nkx2.1-expression progenitors located. We reasoned that Tcf4 might play a role in the regulation of Nkx2.1 progenitors. When we conditionally depleted Tcf4 in the Nkx2.1 lineage, we found that Tcf4 is essential for regional astrocyte allocation. We have included this information in the introduction. ...

Furthermore, most parts of Introduction are about the astrocyte allocation. Authors should introduce more about TCF4.

2. Relation with Pitt-Hopkins syndrome

...It is particularly interesting to examine the behavioral outcomes in Tcf4 cKO mice. After discussing with the editor, we agree that the behavioral tests may extend beyond the scope of the current study. However, we tempered our conclusions related to

the physiological aspects of Pitt-Hopkins syndrome. ...

Although the behavioral tests are beyond the scope of this study, authors should show the direction of future studies including behavioral tests and possible association with Pitt-Hopkins syndrome in Discussion.

3.Methods and Protocols

Was tamoxifen injected only one time at E12.5? Authors should clarified it.

4.Discussion

Again, authors should mention the novelty of this study not only from the view of astrocyte allocation, but from the view of TCF4 function.

Referee #3:

With the additional experiments and reporting in the revised version of their manuscript Zhang et al. have thoroughly addressed my technical concerns and further conceptually strengthened their study, making it now suitable for publication in EMBO J.

Detailed response to editorial and reviewer's comments:

We would like to thank the reviewers for their constructive supports. We are pleased to see that all reviewers are satisfied with the efforts we made during the revision, which have significantly improved our manuscript.

We also thank the editorial team for their time and for highlighting the changes we should make to the manuscript.

Below, we provide a specific response to each point:

Reviewer comments:**Referee #1:**

I have no more question.

Response: we thank the reviewer for his/her support.

Referee #2:

Comments to authors;

The manuscript was partially improved, but some questions I raised have not been responded properly.

Major points

1. Backgrounds

I could not find the parts below which authors suggested in the new version of Introduction.

...Building on our previous work on Tcf4 in the excitatory neuronal lineage (Wang et al., Cereb. Cortex, 2020), we found that Tcf4 is highly expressed in the ventral telencephalon where Nkx2.1-expression progenitors located. We reasoned that Tcf4 might play a role in the regulation of Nkx2.1 progenitors. When we conditionally depleted Tcf4 in the Nkx2.1 lineage, we found that Tcf4 is essential for regional astrocyte allocation. We have included this information in the introduction. ...

Furthermore, most parts of Introduction are about the astrocyte allocation. Authors should introduce more about TCF4.

Response: We thank the reviewer's constructive suggestion. we have incorporated this information in the introduction and provided additional information about Tcf4.

2. Relation with Pitt-Hopkins syndrome

...It is particularly interesting to examine the behavioral outcomes in Tcf4 cKO mice. After discussing with the editor, we agree that the behavioral tests may extend beyond the scope of the current study. However, we tempered our conclusions related to the physiological aspects of Pitt-Hopkins syndrome. ...

Although the behavioral tests are beyond the scope of this study, authors should show the direction of future studies including behavioral tests and possible association with Pitt-Hopkins syndrome in Discussion.

Response: We thank the reviewer for her/his constructive suggestion. We have included a discussion on the future studies regarding mouse behaviors relevant to Pitt-Hopkins syndrome, which we believe is an exciting direction we are pursuing.

3. Methods and Protocols

Was tamoxifen injected only one time at E12.5? Authors should clarified it.

Response: Yes, Tamoxifen was administered in a single dose at E12.5. We have clarified this in the methods section.

4. Discussion

Again, authors should mention the novelty of this study not only from the view of astrocyte allocation, but from the view of TCF4 function.

Response: We included additional information on the potential new functions of Tcf4 in regulating cell fate specification in the discussion section.

Referee #3:

With the additional experiments and reporting in the revised version of their manuscript Zhang et al. have thoroughly addressed my technical concerns and further conceptually strengthened their study, making it now suitable for publication in EMBO J.

Response: we thank the reviewer for his/her support.

All editorial and formatting issues were resolved by the authors.

Dear Yunli,

Congratulations on an excellent manuscript! I am very pleased to inform you that it has been accepted for publication in The EMBO Journal. Thank you very much for your thorough responses to the referee comments and for addressing all editorial requests.

If you have any questions, please do not hesitate to contact the Editorial Office. Thank you for your contribution to The EMBO Journal. Working with you has been a pleasure!

Best regards,

Ioannis
